# Effects of hydrogeochemistry on the microbial ecology of terrestrial hot springs

Carla Barbosa,[1,2] Javier Tamayo-Leiva,[3,4,5] Jaime Alcorta,[3,5] Oscar Salgado,[3,6] Linda Daniele,[1,2] Diego Morata,[1,2] Beatríz Díez[3,4,5]

**ABSTRACT**    Temperature, pH, and hydrochemistry of terrestrial hot springs play a critical role in shaping thermal microbial communities. However, the interactions of biotic and abiotic factors at this terrestrial-aquatic interface are still not well understood on a global scale, and the question of how underground events influence microbial communities remains open. To answer this, 11 new samples obtained from the El Tatio geothermal field were analyzed by 16S rRNA amplicon sequencing (V4 region), along with 191 samples from previous publications obtained from the Taupo Volcanic Zone, the Yellowstone Plateau Volcanic Field, and the Eastern Tibetan Plateau, with their temperature, pH, and major ion concentration. Microbial alpha diversity was lower in acid-sulfate waters, and no significant correlations were found with temperature. However, moderate correlations were observed between chemical parameters such as pH (mostly constrained to temperatures below 70°C), $SO_4^{2-}$ and abundances of members of the phyla Armatimonadota, Deinococcota, Chloroflexota, Campilobacterota, and Thermoplasmatota. pH and $SO_4^{2-}$ gradients were explained by phase separation of sulfur-rich hydrothermal fluids and oxidation of reduced sulfur in the steam phase, which were identified as key processes shaping these communities. Ordination and permutational analysis of variance showed that temperature, pH, and major element hydrochemistry explain only 24% of the microbial community structure. Therefore, most of the variance remained unexplained, suggesting that other environmental or biotic factors are also involved and highlighting the environmental complexity of the ecosystem and its great potential to test niche theory ecological associated questions.

**IMPORTANCE**    This is the first approach to investigate whether geothermal processes could have an influence on the ecology of thermal microbial communities on a global scale. In addition to temperature and pH, microbial communities are structured by sulfate concentrations, which depends on the tectono-magmatic settings (such as the depth of magmatic chambers) and the local settings (such as the availability of a confining layer separating NaCl waters from steam after phase separation) and the possibility of mixing with more diluted fluids. Comparison of microbial communities from different geothermal areas by homogeneous sequence processing showed that no significant geographic distance decay was detected on the microbial communities according to Bray-Curtis, Jaccard, unweighted, and weighted Unifrac similarity/dissimilarity indices. Instead, an ancient potential divergence in the same taxonomic groups is suggested between globally distant thermal zones.

**KEYWORDS**    16S rRNA gene, geothermal field, alpha and beta diversity, thermal water, co-occurrence networking

Geothermal regions are globally distributed in zones of elevated crustal heat flow, preferentially concentrated in areas of active magmatism and/or crustal thinning (1, 2). Crustal heat transfer is locally enhanced by circulating water, which can reach

Address correspondence to Beatríz Díez, bdiez@bio.puc.cl.

Carla Barbosa and Javier Tamayo-Leiva contributed equally to this article. Carla Barbosa is listed first for being, together with the corresponging author, the main researcher involved in the initial formulation, development of analyses, discussion of results, and original writing of the manuscript, while Javier Tamayo was primarily involved in the statistical and ecological analyses of the paper and discussion of results.

The authors declare no conflict of interest.

See the funding table on p. 19.

the surface and form geothermal features. The chemistry of these waters represents an opportunity to explore the processes that occur before the fluids reach the surface. These fluids form primarily from meteoric water and/or seawater (3, 4) that is heated in the upper first few kilometers of the Earth's crust. Aquifer fluids may undergo compositional changes by depressurization boiling, phase separation, mixing, precipitation, or dissolution of minerals or by contributions of magmatic fluids (3, 5–7).

According to pH and major ion concentrations, hot springs are usually classified into (i) NaCl waters, (ii) acid-sulfate waters, and (iii) $HCO_3$ or $CO_2$-rich waters (3, 6–12). NaCl waters are distinguished by their near-neutral pH and high concentrations of $Cl^-$. In contrast, acid-sulfate waters are characterized by high concentrations of $SO_4^{2-}$ and low pH values. The acidity is caused by contributions of HCl and $SO_2$ derived from magmatic/volcanic sources or by the joint action of phase separation and mixing of geothermal fluids (9, 13). In the latter, reservoir fluids boil during ascent and separate into an $H_2S$-$CO_2$-rich steam phase and a liquid-saline phase. The steam phase rises and mixes with shallow or surface groundwater, and the available $H_2S$ is oxidized to form $SO_4^{2-}$. According to geothermal reservoirs conceptual models (2), the third type of water consists of neutral to alkaline pH bicarbonate-rich waters, preferentially located at the periphery of geothermal systems. These waters may originate from the boiling of a $CO_2$-rich geothermal fluid, followed by condensation of the steam phase and fluid interaction with the surrounding rocks.

These aqueous environments provided by hydrothermal activity are colonized by microorganisms that take advantage of specific local hydrochemistry through various metabolic pathways (14, 15). Complex interactions between the local environmental conditions of the hot springs and their microbial communities have been previously described. It has been suggested that temperature would be the factor that modulates the microbial communities inhabiting hot springs in the Tibetan Plateau (16–18) and Malaysia (19). In contrast, pH has been found to be the most critical parameter structuring microbial life in the Taupo Volcanic Zone (20), and its contribution along with temperature has been shown to strongly influence microbial beta diversity in hot springs in Yellowstone National Park (21) and Costa Rica (22). In addition to these factors, other chemical parameters such as total and dissolved organic carbon, redox potential, dissolved sulfide, and elemental sulfur have also been shown to impact specific taxa in thermal communities (18, 23).

However, on a global scale, much less has been studied overall on how thermal microbial communities are structured. Novel and previous data obtained for geothermal areas in Argentina and Malaysia, New Zealand, India, China, Russia, and USA were analyzed together, and no significant effect of temperature and pH on microbial communities was demonstrated, possibly explained by the effect of other multiple factors involved (24). Among these other factors, the hydrochemistry of thermal features has been suggested to influence the community due to varied physicochemical drivers (25, 26), leading to the question of how geothermal processes such as water-rock interactions, boiling, or water mixing affect microbial communities in terrestrial hot spring environments. Seeking to answer this question, chemosynthetic hot spring communities in Yellowstone National Park have been studied, from which it has been proposed that water acidification generated by phase separation controls the availability of nutrients, which influences the microbial ecology of thermophilic communities (27). It has also been proposed that subsurface and near-surface water mixing plays a key role in promoting chemosynthetic biodiversity in geothermal systems (28). Similarly, Guo et al. (16) investigated the role of hydrogeochemical processes on the microbial ecology of hot springs in Yunnan Province (China). The authors proposed that shallow circulation and sulfur oxidation accounted for the hydrochemistry of moderate-temperature acidic springs while high-temperature alkaline springs hydrochemistry was the result of deeper fluid circulation. This distinction was recognized as the main contributor to the environmental variations modulating microbial communities in Yunnan.

In the present study, we investigated the influence of hydrothermal processes on the microbial communities inhabiting hot springs at four emblematic geothermal sites worldwide. We carried out this identification by comparing the hydrochemistry and respective microbial communities of 152 hot springs from the Taupo Volcanic Zone (TVZ; New Zealand), 25 from the Yellowstone Plateau Volcanic Field (YPVF; USA), 11 from the Altiplano-Puna Volcanic Complex (APVC; Chile), and 14 from the Eastern Tibetan Plateau Geothermal Belt (ETPGB; China).

## RESULTS AND DISCUSSION

### Study zones

#### Taupo Volcanic Zone

The Taupo Volcanic Zone (TVZ) (Fig. 1A) is an active zone of calc-alkaline volcanism and intra-arc rifting derived from westward subduction of the Pacific Plate beneath the North Island (29), with voluminous rhyolitic volcanism active since 1.6 Ma ago (30). Rhyolitic pyroclasts and lava flow, mainly discharged by caldera-type volcanoes, and Quaternary sediments cover the area (31). Structures are NNE-trending faults that extended from Mt.

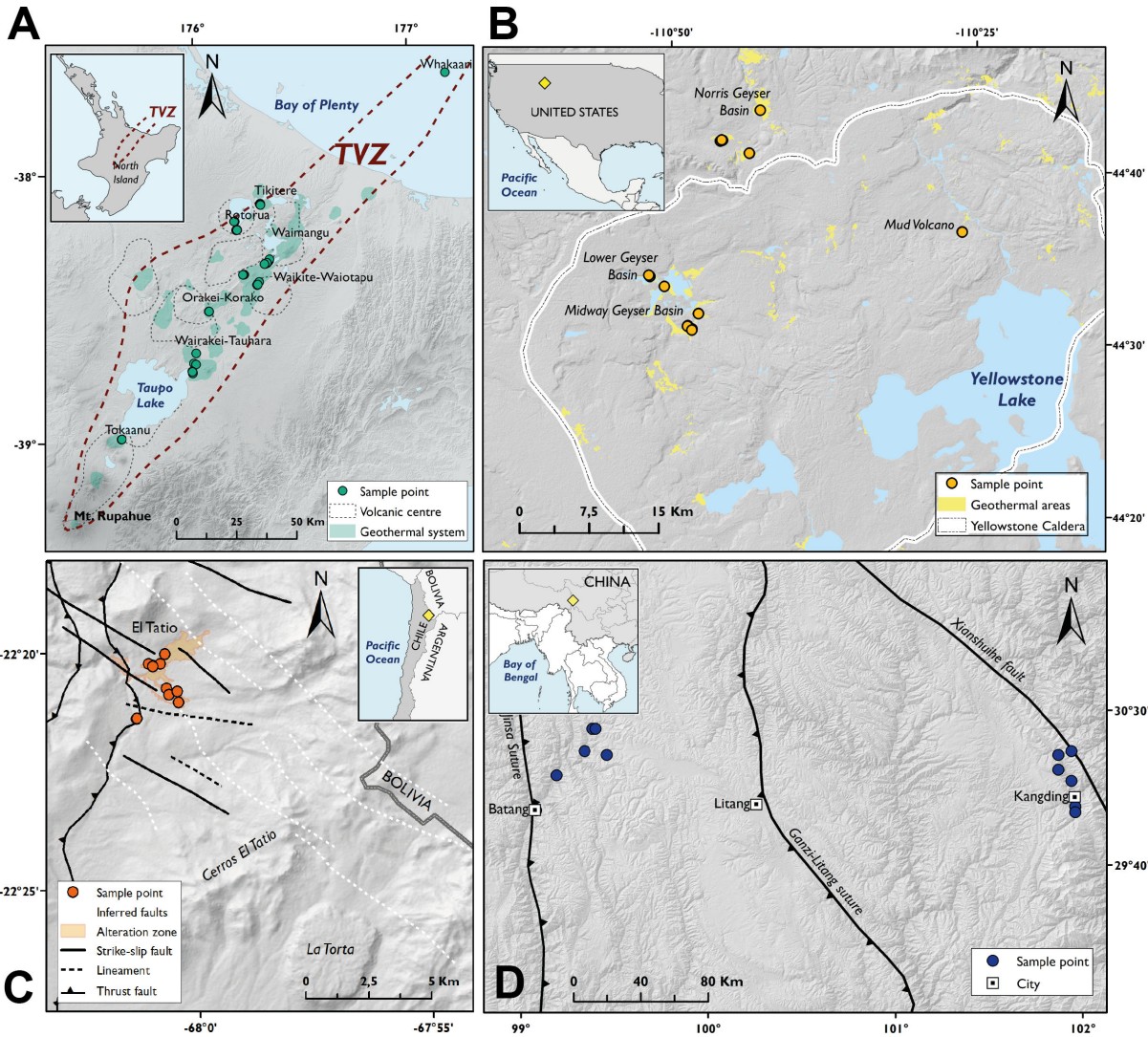

**FIG 1** Location of the analyzed hot springs. (A) Taupo Volcanic Zone (TVZ), (B) Yellowstone Plateau Volcanic Field, (C) El Tatio at the Altiplano-Puna Volcanic Complex, and (D) Eastern Tibetan Plateau Geothermal Belt. Map elaborated with Esri "World Countries Generalized" basemap. Scale not given. 8 June 2023. https://hub.arcgis.com/datasets/esri::world-countries-generalized/explore (accessed 17 August 2023).

Ruapehu to the Bay of Plenty with a 15- to 20-km-wide rift zone, known as Taupo Fault Belt or Taupo Rift (32).

The geothermal heat flux is released mainly in its central part by 23 high-temperature (>250°C) geothermal fields (33) with fluid chemistry greatly varying between and within geothermal systems. Deep geothermal fluids, which reach over 300°C (34), consist of water, carbon dioxide, and chloride as their most important components (35). This water is predominantly of meteoric origin with proportions of magmatic inputs.

### Yellowstone Plateau Volcanic Field

The YPVF (Fig. 1B), located in Yellowstone National Park (USA), has been formed over the past 2.1 Ma due to intraplate hotspot activity. Three cataclysmic volcanic eruptions have been reported in that period (36) producing mainly rhyolitic lava flows, sequences of ash-flow tuffs, and basaltic lava flows without intermediate-composition rocks (37). The latter event, dated to 0.64 Ma, gave rise to the Yellowstone Caldera (36).

The YPVF hosts the largest geothermal region in the world, which comprises more than 10,000 features, including thermal pools, mud pots, fumaroles, and frying pan (38). Its surficial manifestations are found primarily within or near to the margin of the Yellowstone Caldera or along north-south-trending faults outside the caldera (39). Geothermal manifestations vary in gas and water composition, having pH values between 1.5 and 10. From hot springs' hydrochemical analyses, deep fluids are known to reach 340°C to 370°C, which ascend through successively shallower and colder reservoirs (40) where changes in physicochemical conditions lead to water-gas-rock chemical reactions (41). Mixing water has been also described to produce intermediate-composition manifestations (9, 42).

### El Tatio (at Altiplano-Puna Volcanic Complex)

The Andes is an orogenic belt formed due to the subduction of the oceanic Nazca plate under the continental upper South America plate. The APVC, located in the Central Volcanic Zone of the Andes, is one of the largest silicic volcanic fields in the world (43), formed since the Miocene (44). Quaternary volcanism, active fractures, and fault systems in this zone have given the natural conditions to develop high-temperature geothermal systems (45, 46), such as El Tatio.

El Tatio (Fig. 1C) is located over 4,200 meters above sea level, where extreme atmospheric conditions are due to low precipitation and high evaporation rates, high daily temperature oscillations (47), and high rates of ultraviolet radiation (48). The geothermal area encloses about 200 thermal features including geysers, fumaroles, hot springs, and mud volcanoes (49, 50) and extensive sinter deposits. The temperature of the thermal fluids can reach 86°C at the surface, which is the boiling point at this altitude, while the pH ranges from 5.5 to 8.2. Conceptual implications of the system have been deduced: meteoric water recharge at higher altitudes, ~15 to 20 km east of El Tatio (51). Two local reservoirs, one deeper and hotter (260–270°C) and one shallower and cooler (160–170°C), accumulate hot fluids and feed thermal features. In addition, a larger-scale model has been proposed (50), which integrates El Tatio into a regional geothermal system that also comprises La Torta dome 10 km southeast of El Tatio. In this model, a third hotter reservoir is proposed below La Torta dome, from which fluids flow sub-horizontally across NW-striking faults and rise at the intersection between the N- and NW-striking faults found in El Tatio (Fig. 1C).

### Eastern Tibetan Plateau Geothermal Belt

The Tibetan-Himalayan Plateau has been formed by the collision between the Indian and Eurasian tectonic plates since the early Cenozoic (52). The Tibetan Plateau is the largest plateau on Earth (53) and is composed of tectonic terranes accreted to the southern margin of Asia throughout the Phanerozoic (54). In its eastern region, ~4,000 m above sea level is located the Songpan-Ganzi complex, a triangular fold belt containing

multiple lithospheric-scale strike-slip and thrust fault zones (55). This area is covered by granitic rocks, and it is bounded to the west by the Yidun arc across the Ganzi Suture (56).

In the ETPGB (Fig. 1D), about 250 thermal manifestations are distributed in three geothermal belts, each spatially related to a fault system (57). From west to east, (i) the Dege-Batang-Xiangcheng geothermal belt is related to the Dege-Xiangcheng fault system; (ii) the Ganzi-Xinlong-Litang geothermal belt ,to the Ganzi-Litang fault zone; and (iii) the Luhuo-Daofu-Kangding geothermal belt, to the Xianshuihe fault zone (58). It is believed that reservoir fluid upwelling is favored by the permeability given by these fault systems (57). Tang et al. (55, 56) proposed two types of geothermal systems. The Kangding type is characterized by a mantle-derived heat source from the radioactive decay of Cenozoic granite. The main reservoirs are granitic intrusive bodies and Triassic sandstones. On the other hand, the Batang-type systems have a crust-derived heat source that heats Mesozoic intrusive bodies. In both types of system, the recharge has a meteoric origin.

According to the criteria applied in the literature search, nine geothermal fields in the TVZ were included in this study. From south to north, these fields were Tokaanu, Wairakei-Tahura, Orakei-Korako, Waikite, Waiotapu, Waimangu, Roturua, Tikitere, and Whakaari (White Island) (Fig. 1A). Likewise, the fields included in the ETPGB were Batang and Kanding (Fig. 1D), the one included in the APVC was El Tatio (Fig. 1C), and the YPVF geothermal system, which is called Yellowstone thereafter (Fig. 1B). The four study zones are placed in different tectonic settings, which have fundamental implications for the characteristics of their geothermal systems. Some of them are their thermal regime, heat flow, hydrogeologic regime, fluid dynamics, faults and fractures, stress regime, and lithological sequences. All these properties also influence their fluid chemistry as well as local conditions (2). This permits the detection of thermal manifestations of very different compositions in the same geothermal field and, at the same time, surface manifestations of different geothermal fields with similar chemical compositions.

## Hot spring hydrochemistry

Physicochemical parameters and major ion concentrations of the analyzed hot springs covered a broad hydrochemical spectrum (Table S1), which opened the possibility of analyzing microbial communities in different hydrochemical scenarios. Temperatures ranged between 31.5°C and 99°C, pH between 1.5 and 9.9, and electrical conductivity between 236 and 21,000 μS/cm (Fig. 2A through C). Ionic concentrations varied as much as 5 orders of magnitude, such as $Cl^-$, which ranged from 0.05 mg/L to 7,061 mg/L. $SO_4^{2-}$ concentrations ranged from 1.6 mg/L to 2,418 mg/L, and $HCO_3^-$ values ranged from 0 to 1,228 mg/L. As for cationic concentrations, $Na^+$ values reported the broadest range among the samples ranging from 1.7 mg/L and 4,580 mg/L, followed by $K^+$ with concentrations between 1.6 mg/L and 508 mg/L. $Ca^{2+}$ and $Mg^{2+}$ values ranged from 0.36 mg/L to 381 mg/L and from 0 to 173 mg/L, respectively, and Si concentrations ranged from 14.06 mg/L to 368 mg/L. Zonal dependence was observed for the hydrochemical ranges, with greater homogeneity in El Tatio and ETPGB samples than in YPVF and TVZ, although ETPGB samples covered a wide temperature range, between 37.4°C and 88.2°C.

Samples were plotted on a Piper diagram (59; Fig. 2D) according to their major ion concentration. $Cl^-$ was the dominant anion in 71% of the samples, and $Na^+$ was the dominant cation in 92%, meaning that most of the waters were classified as Na-Cl type. Looking at the scale of the study zone, it was noticed that waters in the APVC were all Na-Cl type, while in the ETPGB, there were only $HCO_3$-type waters. Furthermore, most of the Na-$HCO_3$ and Ca-$HCO_3$ samples in the data set belonged to the ETPGB. As for the YPVF and the TVZ samples, they comprised Na-Cl and Na-$SO_4$ waters and a reduced number of Na-$HCO_3$.

### NaCl waters

From the 202 analyzed hot spring samples, 107 were classified as NaCl waters (Fig. 3). $Cl^-$ concentrations at El Tatio were the highest of all samples, reaching 7,062 mg/L,

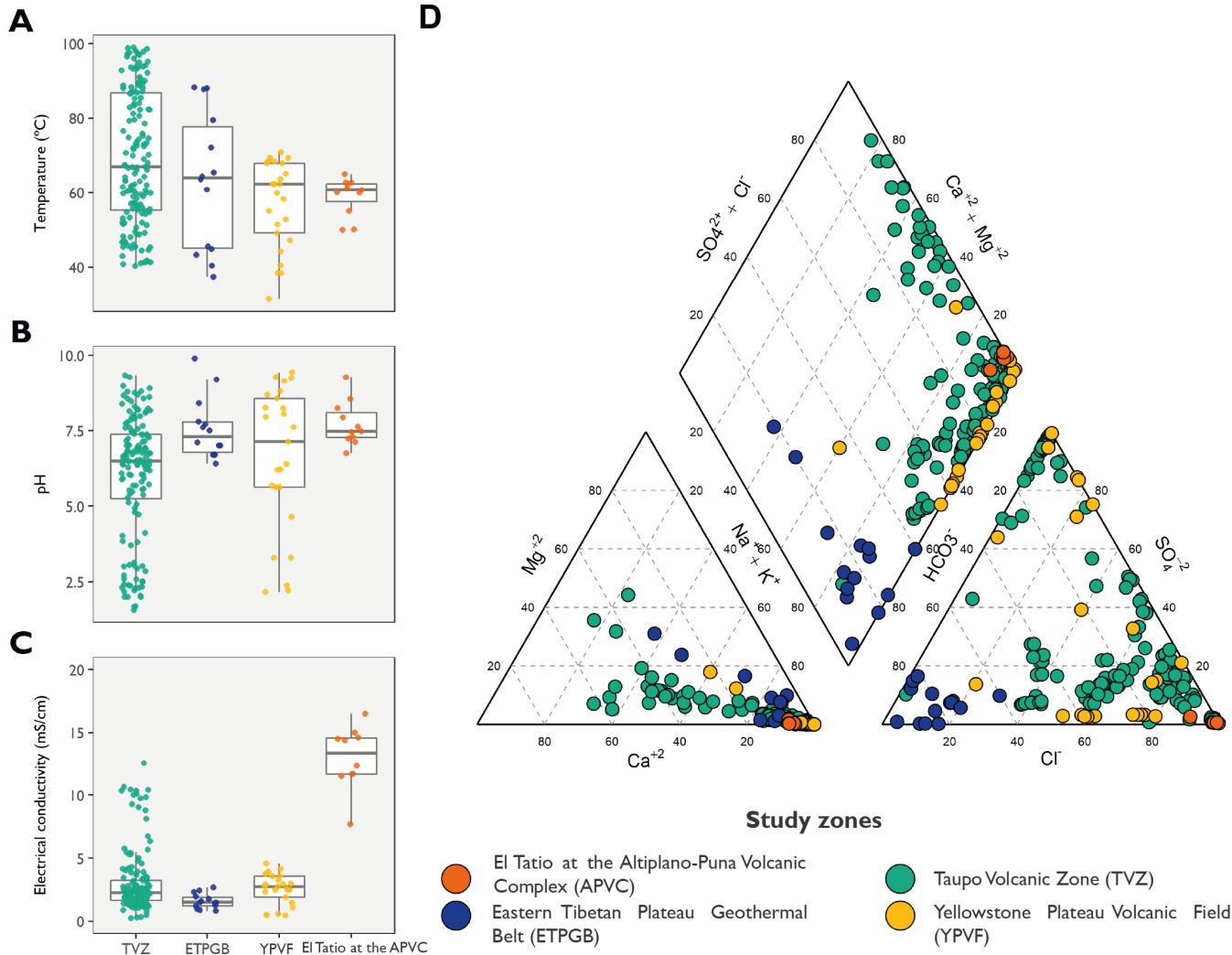

**FIG 2** Distribution of temperature (A), pH (B), and electrical conductivity (C) of the samples at each study zone. (D) Piper diagram of the analyzed samples by study zone.

while Tokaanu's Cl⁻ concentrations reached 3,234 mg/L, being the highest in the TVZ. High salinities in Tokaanu have been interpreted as a cause of steam loss and surface evaporation processes (60–62). Also, both processes have been described as two of the most important secondary events at El Tatio (47, 51), which probably also contribute to concentrate the Cl⁻ and many other compounds in the rising water. In contrast, NaCl waters from Wairakei-Tauhara, Waimangu, Rotorua, Orakei-Korako, and Yellowstone had Cl⁻ concentrations below 1,314 mg/L and higher ratios of $HCO_3^-$ to Cl⁻ and $SO_4^{2-}$ to Cl⁻, which may suggest that other secondary processes are controlling their major element chemistry.

## Acid-sulfate waters

Of the 60 acid-sulfate water samples (Fig. 3), 3 of them were taken at the active White Island volcano. The three analyzed samples had pH values between 3.05 and 5.41, and $SO_4^{2-}$ concentrations, between 804 and 1,969 mg/L. Fluid sources identified include meteoric water, seawater, and magmatic steam (63–65). The latter has been shown to contribute sulfur more as $SO_2$ and less as $H_2S$ (63), and these inputs are responsible for the low pH values reported (66).

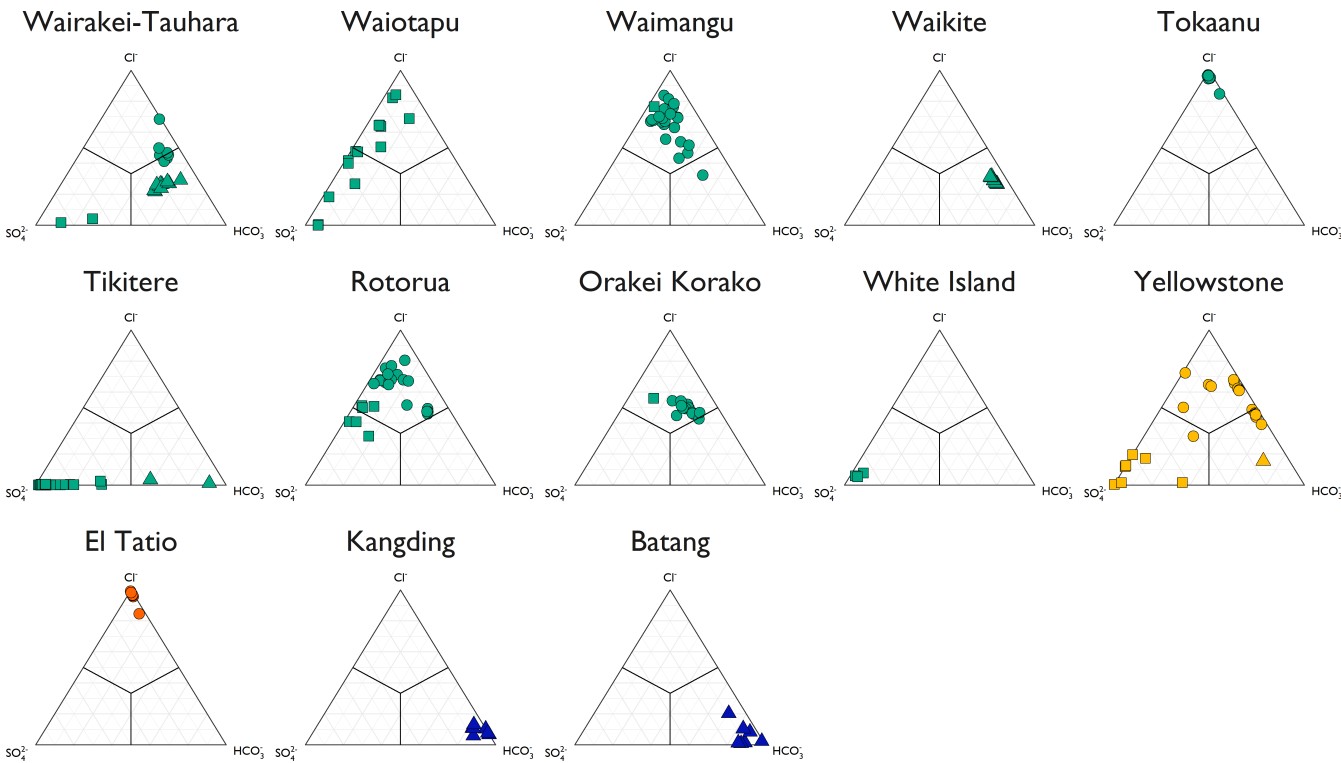

**FIG 3** Ternary SO₄-Cl-HCO₃ diagrams for each geothermal field. Samples are represented according to the geothermal area and water type.

The remaining 57 samples belonged to Yellowstone, Waiotapu, Wairakei-Tauhara, and Tikitere, and their acidity has previously been interpreted as the result of S-rich fluid phase separation and condensation of the steam phase followed, in many cases, by mixing with shallower water (9, 67–70). In addition, Nordstrom et al. (9) deduced many mixing processes in the YPVF that can explain the $SO_4^{2-}$ and $Cl^-$ concentrations in the analyzed waters (Fig. S1).

Hydrochemical trends were observed at Tikitere and Waiotapu, where samples displayed between the $SO_4^{2-}$ and $HCO_3^-$ vertex of the ternary diagram in Tikitere and the $SO_4^{2-}$ and $Cl^-$ vertex in Waiotapu (Fig. 3). These samples exhibited pH values of 1.57 to 6.48 and 2.31 to 5.55, respectively. Lower pH values were observed to be correlated with higher Eh values (Fig. S2), leading to relate acidity to shallower fluid circulation. High concentrations of sulfate in these acidic samples also contribute to consider oxidation of reduced S as an important mechanism of acidity. More information on water chemistry would be needed to evaluate the role of water-rock interactions in the formation of these acid-sulfate waters.

### $HCO_3^-$ or $CO_2$-rich waters

For Batang, Kangding, Waikite, Waimangu, Tikitere, Wairakei-Tauhara, and Yellowstone, 35 samples were classified as $HCO_3^-$ or $CO_2$-rich water. The highest $HCO_3^-$ concentrations were found in the ETPGB (412 mg/L > $HCO_3^-$ > 1,228 mg/L). Ratios of $Na^+$ to $Cl^-$ in these samples were above unity, suggesting that $Na^+$ originated from the dissolution of silicate minerals (71).

In the TVZ, Waikite bicarbonate waters have been interpreted as peripheral waters, being an outflow from the Waiotapu geothermal field (67, 72). In that case, bicarbonate may have originated from the interaction of $CO_2$ with the surrounding rocks. This same origin is believed to have the bicarbonate-rich waters of Wairakei-Tauhara.

## Microbial communities

The 16S rRNA gene is a universal genetic marker for Bacteria and Archaea that allows us to compare the microbial structure of the communities between hot springs and analyze the biological response against the metadata of interest. Standardized sequence processing was performed on the 11 samples from El Tatio and the 191 pre-existing samples in the National Center for Biotechnology Information (NCBI) Sequence Read Archive (SRA) database from the other three study zones. A total of 7,505,257 clean sequences were recovered from the common V4 region of the 16S rRNA gene across the 202 hot spring samples. These sequences represented 950 archaea and 16,398 bacteria amplicon sequence variants (ASVs). Among the clean sequences, 6,348,970 belonged to the TVZ (median = 26,467), 843,032 to the YPVF (median = 32,110), 360,291 to the ETPGB (median = 26,041), and 256,760 to the APVC (median = 24,063; Table 1).

### Alpha and beta diversity

Rarefaction curves analysis showed that ASV saturation was reached beyond 15,000 reads per sample (Fig. S3). Analysis of microbial community diversity in each hot spring showed a wide range of observed richness between 9 and 904 ASVs, while the Shannon index ranged from 0.14 to 5.6 (Table S2). Diversity values were calculated by habitat (water or mat), study zone (4), and geothermal field (13). No statistically significant differences in observed richness ($P > 0.05$) were found grouping by habitat or study zone. However, the Shannon index showed significantly ($P < 0.05$) lower diversity in water communities (2.60 in the ETPGB and 2.77 in the TVZ) compared to microbial mat communities (3.24 in El Tatio and 3.54 in the YPVF, Fig. S4). In contrast, when looking at the scale of the geothermal field, a greater contrast was observed in both metrics. The highest Shannon indexes (3.5–3.9) were found at Wairakei-Tahura (TVZ), Waikite (TVZ), and Yellowstone (YPVF), which also had the highest observed richness along with Batang (ETPGB) and Waimangu (TVZ; 118–200). In contrast, the least diverse communities were found at Tikitere (TVZ), Waiotapu (TVZ), and White Island (TVZ; 1.3–2.3; 23–42).

To further analyze the influence of water chemistry on alpha diversity, correlations between chemical parameters and diversity indexes were calculated for each study zone (Fig. S5). No statistically significant correlations were found between temperature and alpha diversity in any study zone, which supports the findings of Hamilton et al. (73) and Power et al. (20) but contrasts with two previous intercontinental studies that correlated hot spring temperature with alpha diversity (74, 75). On the other hand, observed richness and Shannon index correlated with pH ($R = 0.49, 0.3; P < 0.05$) and $SO_4^{2-}$ concentrations ($R = -0.51, -0.34; P < 0.05$) in the TVZ microbial communities, which was already detected by Power et al. (20). The authors also showed that this effect of pH on alpha diversity was constrained to specific temperature ranges. Thus, samples from

**TABLE 1** Number of 16S rRNA amplicon sequences from each set of samples before and after filtering

| Source | Country | Study zone | Samples | Geothermal systems | Total sequences | Filtered sequences | Median filtered sequences |
|---|---|---|---|---|---|---|---|
| Power et al. (20) | New Zealand | Taupo Volcanic Zone | 152 | Rotorua (RO), Orakei Koraro (OK), Tokaanu (TK), Tikitere (TT), Waiotapu (WP), Waimangu (WG) Waikite (WK), Wairakei-Tahura (WT), White Island (WI) | 9,254,997 | 6,348,970 | 26,467 |
| Hamilton et al. (73) | USA | Yellowstone Plateau Volcanic Field | 25 | Yellowstone (YE) | 1,187,293 | 843,032 | 32,110 |
| Guo et al. (16) | China | Eastern Tibetan Plateau Geothermal Belt | 14 | Batang (BT), Kangding (KD) | 427,442 | 360,291 | 26,041 |
| This paper | Chile | Altiplano Puna Volcanic Complex | 11 | El Tatio (TT) | 389,044 | 256,760 | 24,063 |

all zones were grouped by temperature, which showed that the correlation between pH and alpha diversity was limited to temperatures below 70°C (Fig. S6). This led to the establishment that acid-sulfate waters have lower diversity indexes compared to bicarbonate and NaCl water at temperatures below 70°C (Fig. 4).

To analyze the variation in species composition among different hot springs, geothermal fields, and zones, an ordinate analysis was performed on the weighted Unifrac Beta diversity index. Axis 1 (36.3%) and Axis 2 (14.1%) generated with the multidimensional metric scaling (MDS) analysis explained a total of 50.53% of the variance (Fig. 5). The results did not show a clear distribution of the samples according to their study zone in the ETPGB and TVZ microbial communities. On the contrary, samples from El Tatio showed small phylogenetic distances with most of the YPVF samples.

## Multiple variables as conditioning factors of thermal microbial taxonomic composition

Microbial communities in the TVZ and the ETPGB were mainly composed of Aquificota and Proteobacteria phyla, comprising 55% and 56% of their abundance, respectively (Fig. S7A). More specifically, the dominant families in the TVZ were Aquificaceae, Hydrogenothermaceae, and Hydrogenobaculaceae (Fig. S7B). The two former Aquificota families also dominated in the ETPGB along with the Thermaceae and Comamonadaceae families. In contrast, the El Tatio communities were mainly composed of the phyla Chloroflexota, Bacteroidota, and Cyanobacteria, covering 62% of its relative abundance. The dominant families in this field were Chloroflexaceae, an uncultured member of Armatimonadota, Roseiflexaceae, and Nostocaceae. At El Tatio, a previous study showed that Chloroflexota accounts for the majority of microbial mat and sediment sequences between 38°C and 54°C, whereas Deinococcota dominated at higher temperatures (76). Similarly, Chloroflexota, Bacteroidota, and Cyanobacteria, together with Proteobacteria, comprised 55% of the relative abundance of YPVF communities. The dominant families in the YPVF were Leptococcaceae, Roseiflexaceae, and Thermaceae.

Since temperature and pH have traditionally been considered to have a strong influence on shaping microbial community structures in hot springs, this correlation was tested with the MDS axes in the different samples. The results showed that temperature

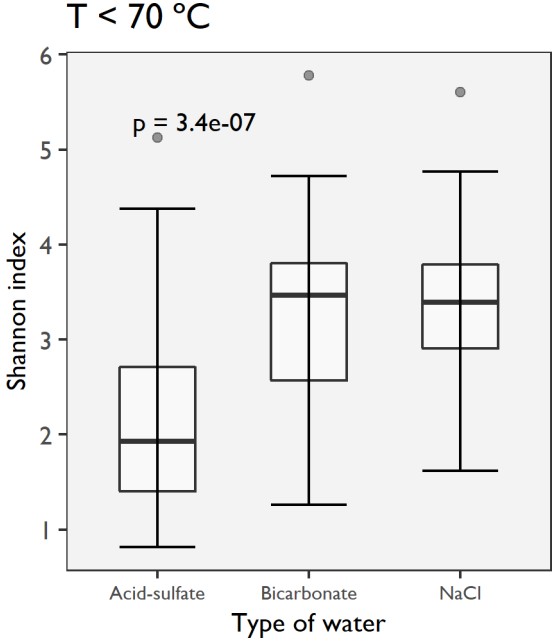

**FIG 4** Shannon index for microbial communities inhabiting acid-sulfate, bicarbonate, and chloride waters at temperature below 70°C.

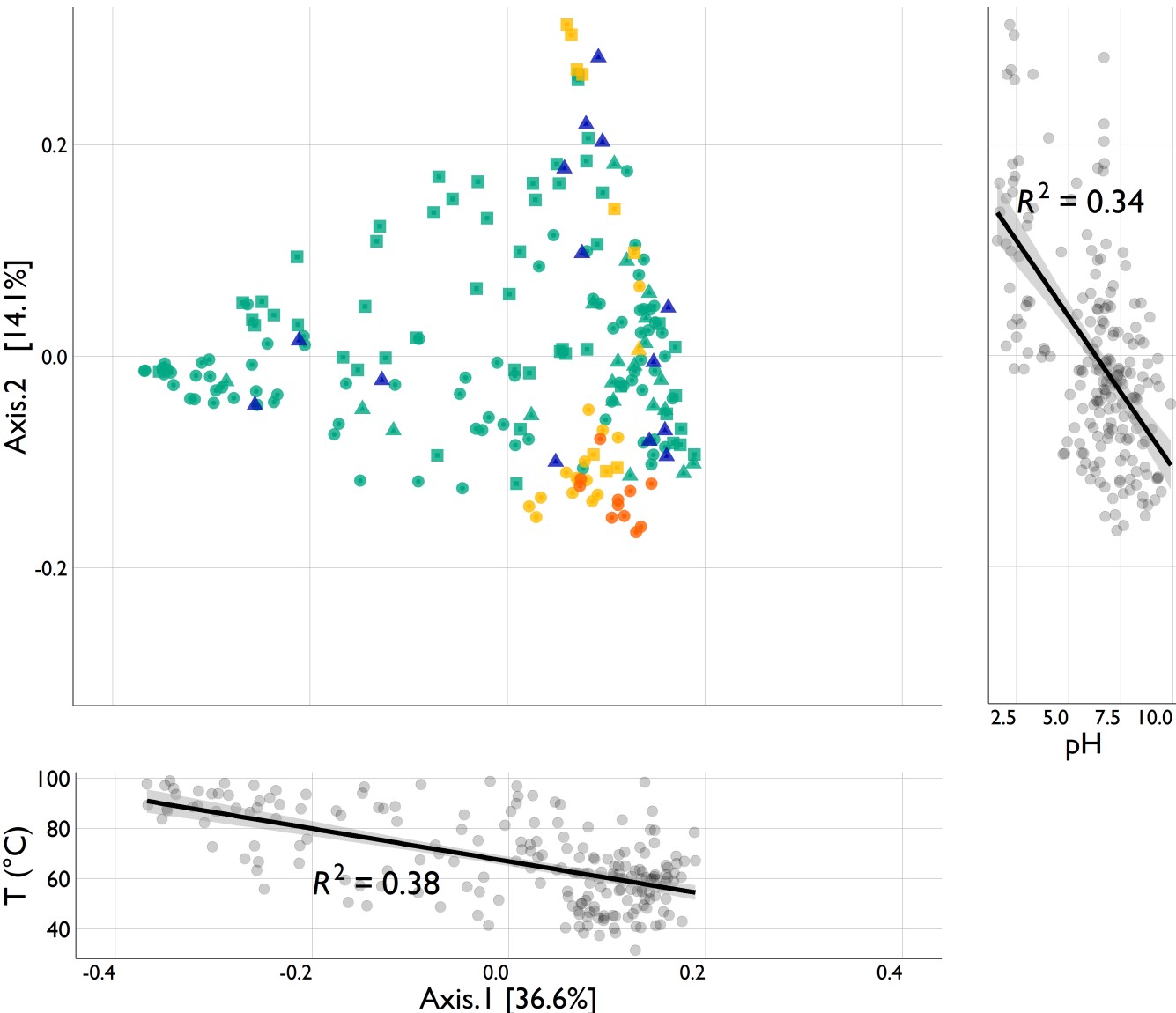

**FIG 5** Axis 1 and Axis 2 resulted from the MDS analysis performed on the weighted Unifrac distances between microbial communities. (Bottom) Correlation of Axis 1 with temperature. (Right) Correlation of Axis 2 with pH.

correlated moderately with Axis 1 ($R^2 = 0.36$, $P < 0.05$), and pH, with Axis 2 ($R^2 = 0.34$, $P < 0.05$) (Fig. 5A and B). Therefore, it is suggested that the structure of the analyzed samples was partially influenced by temperature and pH gradients. No ordering by water type was observed in the distribution of microbial communities, except for communities of the YPVF acid-sulfate waters that were preferentially distributed along Axis 2.

To further assess potential drivers of community composition, temperature, pH, and major ion concentrations were subjected to principal component analysis (PCA) (Fig. S8). The resulting principal components, together with observed richness and microbial community habitat (water or mat), were used as explanatory variables in a permutational analysis of variance (PERMANOVA). Results showed small contributions from each variable, rather than a single principal explainer (Table 2). The highest linear correlation of the hydrochemical factors was 8.3% given by PCA principal component 4 (PC4), which was highly related to temperature and $Mg^{2+}$ and Si concentrations ($P < 0.001$). PC2, mostly associated with pH, $SO_4^{2-}$, and $Mg^{2+}$, as well as PC3 associated with $HCO_3^-$ and Si also showed a significant influence on beta diversity, each explaining less than 5%

**TABLE 2** PERMANOVA results using as explanatory variables observed richness, habitat (water/microbial mat), and the principal components of the PCA constructed with hydrochemical variables[a]

|                   | $R^2$  | F-model | Pr(>F)      |
|-------------------|--------|---------|-------------|
| Observed richness | 0.0212 | 6.37    | $P < 0.001$ |
| Habitat           | 0.0376 | 11.26   | $P < 0.001$ |
| PC1               | 0.0211 | 6.32    | $P < 0.01$  |
| PC2               | 0.0453 | 13.56   | $P < 0.001$ |
| PC3               | 0.0449 | 13.44   | $P < 0.001$ |
| PC4               | 0.0834 | 24.95   | $P < 0.001$ |
| PC5               | 0.0205 | 6.13    | $P < 0.001$ |
| PC6               | 0.0103 | 3.10    | $P < 0.01$  |
| PC7               | 0.0090 | 2.71    | $P < 0.05$  |
| PC8               | 0.0051 | 1.55    | $P > 0.1$   |
| PC9               | 0.0099 | 2.95    | $P < 0.05$  |
| PC10              | 0.0032 | 0.97    | $P > 0.1$   |
| Residual          | 0.6320 |         | -           |
| Total             | 0.944  |         | -           |

[a]Distances were based on weighted Unifrac metric. Pr(>F), p-value associated with the F statistic.

of the total variance ($R^2 = 0.045$, $P < 0.001$). Microbial community habitat contributed to explaining 3.8%, while observed richness explained 2.1%. Hence, 63.19% of the variance remained unexplained, suggesting other environmental variables involved in structuring these communities, such as humidity, solar radiation (48), or other hydrochemical variables not considered in the present study. In the literature, it has been suggested that microbial community variance is better constrained with hydrochemistry where photosynthesis is absent (23, 77), which is consistent with PERMANOVA results at temperatures over 75°C (Table S3). In these cases, the concentrations of dissolved species regulate the metabolic pathways that microorganisms can resort to (77). On the other hand, the influence of the biological component, such as the interaction between taxa, could also play a role in shaping these thermal communities.

Furthermore, when assessing the effect of hydrochemical gradients on dominant taxa (>0.1%), only weak to moderate correlations ($P = 0.2$ to $0.7$; $P < 0.05$) were identified with the relative abundances of these taxa (Fig. 6). Chloroflexota, Bacteroidota, and Cyanobacteria, the most abundant phyla of the YPVF and the APVC, were moderately positively correlated with pH and moderately negatively correlated with $SO_4^{2-}$. These correlations confirm the circumneutral to alkaline pH at which members of Chloroflexota and Cyanobacteria grow optimally in terrestrial hot springs, forming phototrophic mats (e.g., 22, 75, 77–80).

Chloroflexaceae family showed the highest correlation with pH (Fig. S9), as reported for the genus *Chloroflexus* (73). However, the highest correlation of pH and $SO_4^{2-}$ was observed for members of the family Thermaceae (Deinococcota) and a family with uncultured members of Armatimonadota. Power et al. (20) and Hamilton et al. (73) previously recognize this positive correlation between pH and the *Thermus* genus of the Thermaceae family in YPVF and TVZ, respectively. Similarly, thermophilic members of Armatimonadota have only been found at neutral or alkaline pH (81). In contrast, the abundances of Thermoplasmatota and Campilobacterota increased with $SO_4^{2-}$ and decreased with pH, as observed for Desulfurellaceae and a family with uncultured members of the phylum Thermoplasmatota. The family Acidithiobacillaceae (Proteobacteria) was similarly related to the above parameters, being the only family of Proteobacteria correlated with temperature, which are preferentially present at lower temperatures. This family and, in particular, its best-known genus, *Acidithiobacillus*, have been extensively reported in acidic terrestrial hot springs and acid mine drainages (e.g., 82, 83, 84 ).

Members of Aquificota, a dominant phylum in ETPGB, were correlated with different chemical parameters. Aquificaceae were preferentially distributed at high temperature,

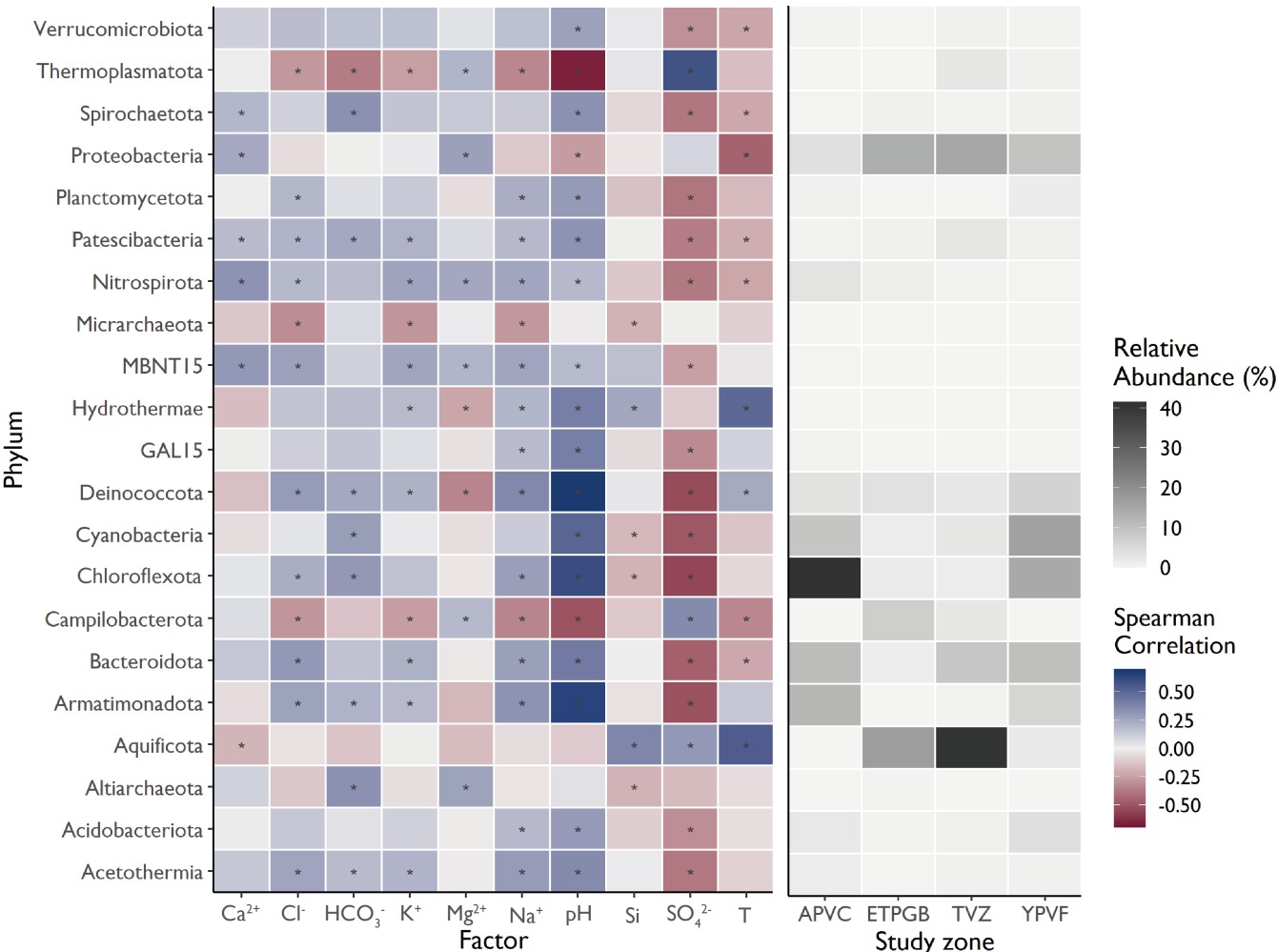

**FIG 6** Highest Spearman correlations (*P* > 0.3; *P* > 0.01) between phylum abundance and hydrochemical variables. Sequences were previously filtered by relative abundance (<0.1%).

as previously reported for the genus *Aquificae* on the Tibetan Plateau, Yellowstone, and Iceland (18, 74, 79). Members of Hydrogenobaculaceae were not affected by temperature but were negatively correlated with pH and $HCO_3$ and positively correlated with $SO_4^{2-}$. The genus *Hydrogenobaculum* has been isolated from sites with pH values between 1.02 and 5.75 and high concentrations of $H_2$, $H_2S$ and $CO_2$ (85). Members of this genus have been found to grow on these chemical species and use them together with thiosulfate as energy sources (85), illustrating their dependence on hot spring chemistry (Fig. S9). In addition, Hydrothermae and Bathyarchaeia (Ca. Bathyarchaeota) also showed a moderately positive correlation with temperature.

Weaker correlations were found between pH, $SO_4^{2-}$, temperature, $Cl^-$, $Na^+$, $Ca^{2+}$, and $Mg^{2+}$ and some other taxa (*P* < 0.4; *P* < 0.05). This again suggests that the action of environmental conditions on microbial communities in thermal environments is not limited to the individual effects of one or a couple of parameters but to a set of complex interactions between biotic and abiotic elements, as described by niche theory (86).

In that sense, many interactions between microorganisms forming photoautotrophic mats have already been described (78, 87, 88). In the present study, we further analyzed biotic interactions among those phyla that were more correlated with water chemistry and performed microbial co-occurrence networks by geothermal zone (Fig. 7). This analysis showed higher modularity and lower betweenness in El Tatio and ETPGB than in YPVF and TVZ. These differences can probably be explained by the difference in the

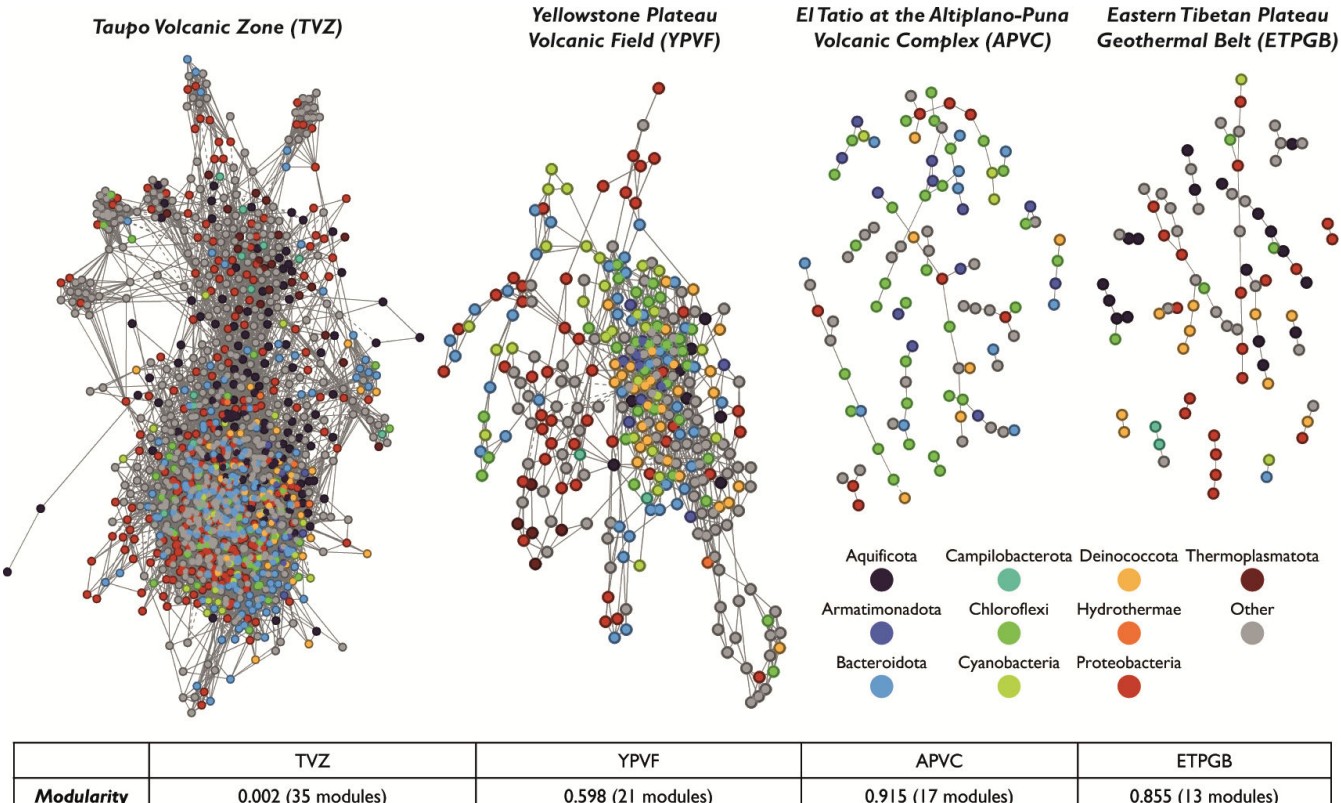

| | TVZ | | YPVF | | APVC | | ETPGB | |
|---|---|---|---|---|---|---|---|---|
| **Modularity** | 0.002 (35 modules) | | 0.598 (21 modules) | | 0.915 (17 modules) | | 0.855 (13 modules) | |
| **Betweenness** | Median | SD | Median | SD | Median | SD | Median | SD |
| | 620.928 | 2,371.486 | 329.479 | 945.164 | 0.500 | 8.400 | 6.000 | 31.082 |

**FIG 7** Microbial co-occurrence network of the four study zones. Nodes (ASVs) are colored by phylum. Colored nodes represent those phyla with the highest correlations with chemical parameters. Positive connections between nodes are displayed with continuous lines and negative interactions with segmented lines.

number of samples analyzed for each zone, which allowed more significant connections in the latter two. In addition, as inferred from their correlations between chemical factors, positive connections were found between some phyla that are affected by the same environmental factors. This was observed in the Armatimonadota and Chloroflexota nodes and in the Cyanobacteria, Chloroflexota, and Bacteroidota nodes, which correlated similarly with pH and $SO_4^{2-}$.

### Role of geothermal processes on chemistry and microbial communities

PERMANOVA results indicated that the eigenvectors PC4 (temperature, $Mg^{2+}$, and Si) and PC2 (pH, $SO_4^{2-}$, and $Mg^{2+}$) were the most important explanatory factors of community structure. However, $Mg^{2+}$ and Si concentrations did not show other relevant correlations with the community as did temperature, pH, and $SO_4^{2-}$. In addition, pH and $SO_4^{2-}$ were negatively correlated (Fig. S10), which is probably because the main source of acidity in geothermal waters is the oxidation of $H_2S$ that produces $SO_4^{2-}$ (9). These and other correlations between pH and chemical variables (Fig. S10) make it difficult to determine what the actual influence of each parameter is on the microbial communities. Moreover, many water-rock interaction rates are favored at low pH values, such as the dissolution of Ca-bearing minerals (89) that increase the concentration of $Ca^{2+}$ and $Mg^{2+}$ in the water. This interaction could explain the low to moderate correlations between certain taxa and $Ca^{2+}$ and $Mg^{2+}$ (Fig. 6; Fig. S9).

On the other hand, the availability of reduced sulfur in surface manifestations is linked to regional and local conditions that shape the entire geothermal system, such as its tectono-magmatic setting, and geological and hydrogeological mechanisms (2). The

regional conditions that allow the formation of steam-heated water have been evaluated by Guo et al. (90) in the Tibetan Plateau. These studies explained the absence of acid-sulfate waters in this geothermal zone by (i) the lack of a confining layer separating the underlying NaCl waters from steam after phase separation and (ii) the depth of magmatic chambers that could play a role in the low $H_2S$ concentrations in geothermal fluids. Moreover, it has been shown that microbial communities on the Tibetan Plateau are modulated by temperature (16–18), which might be explained to some extent by the absence of acidic springs. In fact, Guo et al. (16) found and analyzed a thermal spring with pH <5 and demonstrated that its microbial community structure differed from the rest of hot springs in this geothermal zone.

In addition to the mechanisms proposed by Guo et al. (16), local processes also affect the acidity of these steam-heated water, such as the dissolution/precipitation of certain minerals or the ratio of shallow water and S-rich steam in the fluid. Hot springs fed by concentrated $H_2S$-$CO_2$-rich fluid would have higher $SO_4^{2-}$ concentrations and lower pH values than those with a higher shallow component. These local variations can be evidenced in Tikitere, where samples showed a wide pH range. As shown in Fig. S11, 51.9% of the community structure at Tikitere was explained by Axis 1, which was moderately correlated with pH ($R^2 = 0.48$). Thus, the community structure of Tikitere responds to the pH gradient. However, no correlation was found between pH, $SO_4^{2-}$ and alpha diversity in this geothermal field, which could be attributed to the temperature variations experienced by the hot springs (from 44.4°C to 86.8°C), which probably also affect the communities. Additionally, deep geothermal fluids may precipitate alunite, anhydrite, or pyrite at different depths and decrease their sulfur concentrations, as what occurred in Yellowstone (9, 91). This would prevent the formation of acid-sulfate waters (9), which could limit the development of acidophile communities.

In contrast to the mentioned processes, no clear influence of $Na^+$, $K^+$, and $Cl^-$ concentrations was detected on the analyzed microbial communities. Therefore, water-rock interactions that incorporate $Na^+$, $K^+$, and $Cl^-$ to the water or secondary processes that concentrate them do not appear to play a critical role in modeling alpha diversity, community structure, or the taxonomy of thermophilic microbial communities, but rather, there are other environmental factors that shape these communities, which might occur in El Tatio and Tokaanu geothermal fields.

### Influence of geographic distance on microbial community's beta diversity

To evaluate whether geographic distance influences the similarity among microbial communities, its effect was determined at three different spatial scales defined as local, regional, and global. The local scale includes samples within the same geothermal system (< 58 km), while the regional scale defines the maximum distance between samples from the same study zone (<279 km). Finally, the global scale comprises distances between different study zones (<8,576 km). As there are no data between 279 km and 8,576 km, this scale was defined as a continental scale.

At the local scale, all beta diversity indexes showed significant ($P < 0.01$) negative relationships with distance (Fig. 8). Jaccard and Bray-Curtis indexes showed weak correlations with distance ($R^2 = 0.11$, $P < 0.01$) but still higher than phylogeny-based relationships ($R^2 <0.01$, $P < 0.01$). At the regional scale, only unweighted Unifrac similarities were significant ($P < 0.01$), but with almost zero correlation with geographic distance ($R^2 = 0.003$). Similarly, on a global scale, unweighted and weighted Unifrac showed almost absent correlations with geographic distance ($R^2 = 0.0033$, $R^2 = 0.013$, $P < 0.01$). From these results, no distance decay on microbial communities' similarity was inferred based on Jaccard and Bray-Curtis indexes or unweighted and weighted Unifrac distances. Thus, it is shown that geography does not explain the change in community composition across hydrothermal regions, at least for the four indices tested here. However, as the microbial communities tested were taxonomically related on a global scale, this might suggest for a potential ancient divergence in the same taxonomic groups between globally distant thermal zones, and thus, with the time, some endemic

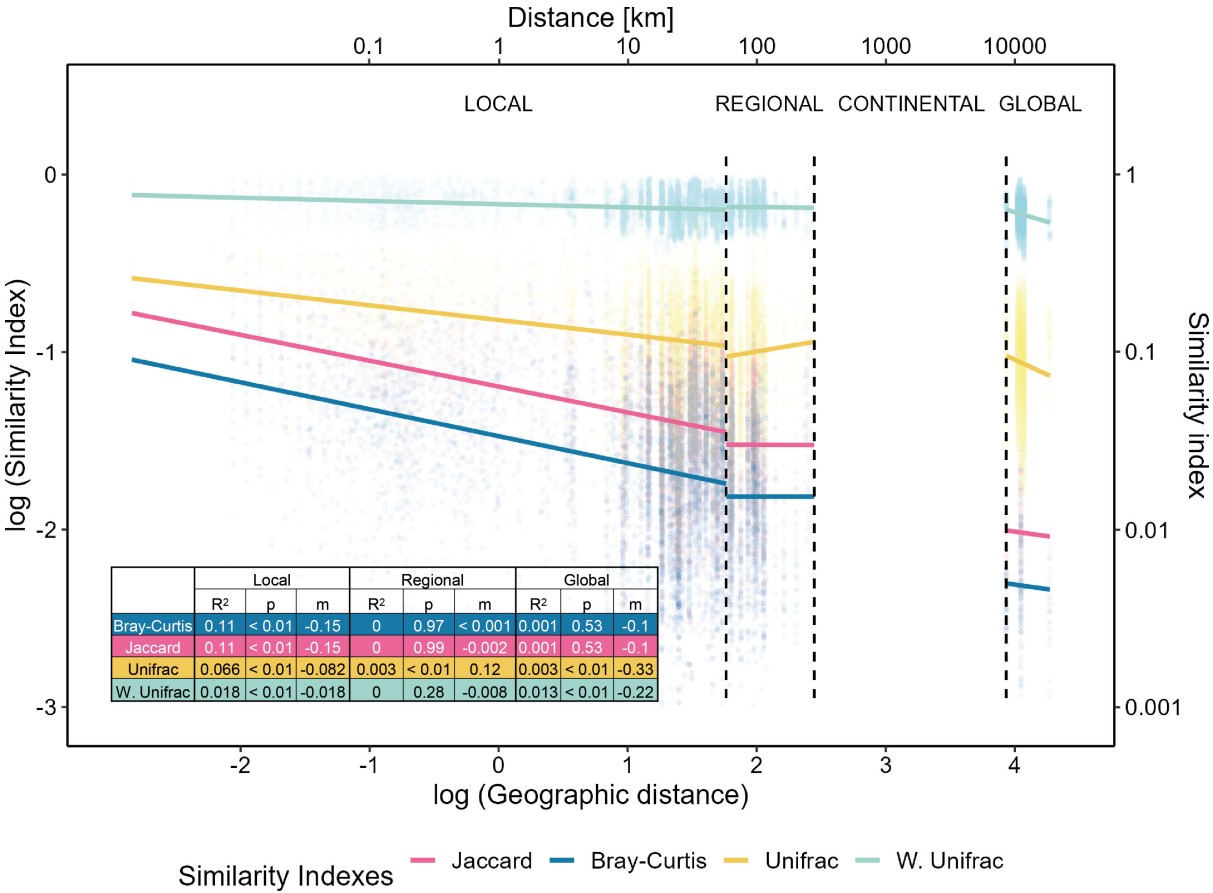

**FIG 8** Jaccard, Bray-Curtis indexes, and unweighted Unifrac and weighted Unifrac distances of the microbial communities according to their geographic distance.

character in each of them appeared as previously reported in the literature for some specific taxa (92–96).

## Considerations for comparative analyses of geothermal zones

It is worth mentioning that the four different sampling zones and the methodologies used in each may present some drawbacks when comparing them, regardless of the fact that studies and samples with significant metadata and the same V4 16S rRNA amplified region were selected for this study. *In situ* temperature and pH were measured with equipment having different accuracies, which could lead to a bias. Similarly, microbial community sampling points varied with respect to where overlying water of the hot springs was sampled (which are not even mentioned in some of the studies). As can be seen in Table S4, anion and cation quantification methodologies varied between studies. Despite this, this fact should not be a limitation for the comparison of major elements, which are mostly present in high concentrations compared to the detection limits and sensitivity values of the equipment. However, although the sequencing effort was normalized by rarefaction analysis, there was a bias of the different primers targeting the same region, e.g., 515F and 806R for V4, by different modifications for ambiguous bases showing different percentages of potential amplification against the NR SSU silva 138.1 (from 9.3% to 83.9%). While some phyla showed similar bias with all primer sets (e.g., Thermoplasmatota, Aquificota, and Deinococcota), others did not (e.g., Bacteroidota, Chloroflexi, and Proteobacteria). The latter showed that the primers used in the present study (515F and 926R) currently cover larger number of taxa (83.9%) of the microbial

communities than other primers sets, highlighting that further studies will be necessary in the future for the design of better primers (97).

## Conclusions

Terrestrial hot springs are a valuable natural resource that host an exceptional biotic diversity. This study attempted to compare the microbial diversity and hydrochemistry of 202 hot springs from four geothermal areas of the world. The comparison of data from different studies revealed multiple biases, mainly associated with the methodologies of analytical chemistry, nucleic acid extraction, and amplification. Despite these limitations, statistical analysis showed that 24% of the variance in thermal community structure was explained by temperature, pH, and hydrochemistry of major elements. This highlights the relevance of these parameters in the microbial ecology of thermophilic communities and the potential influence of geochemical processes on them. At the same time, hydrogeochemistry was consistent with the types of underlying geological processes, showing their influence on the assembly of microbial communities. Furthermore, correlations of dominant taxa (such as Chloroflexota, Bacteroidota, Cyanobacteria, Deinococcota, Thermoplasmatota, Aquificota, and Armatimonadota) with hidrogeochemistry showed that these processes also influence their biotic interaction in microbial communities. Meanwhile, these assemblages showed consistency across the globe as their microbes appear to diverge early in the same taxonomic groups between globally distant thermal zones. Altogether, these results indicate the need to develop collaborative and standardized global efforts to better understand the effect of hydrochemistry on microbial communities, as well as to continue to conduct local studies of geothermal fields located in different areas of the globe, taking into account the singularities of each zone and its particular microbial communities.

## MATERIALS AND METHODS

### El Tatio hot spring sampling and analysis

Water and microbial mat samples were obtained from 11 hot springs of the El Tatio geothermal field in January 2020 (Fig. 1C). Temperature and pH were measured *in situ* with a WTW Multi340i multiparameter analyzer (Table S4), with the accuracy of 0.1°C and 0.01, respectively. Water from the hot springs was collected in 125-mL polyethylene bottles previously rinsed with ultra-pure water and $HNO_3$ for cation measurements. Water was filtered through 0.45-µm Millipore membrane filters (Merck; Darmstadt, Germany), except the samples for bicarbonate analysis. Samples for cation measurements were acidified with $HNO_3$ 4 N, and samples for silica quantification were diluted to 10%, vol/vol. All samples were stored at 4°C until analysis. Cation concentrations were obtained by Flame Atomic Absorption Spectroscopy (Perkin Elmer Pinaacle 900F) at the Andean Geothermal Center of Excellence (CEGA) of the Geology Department of the University of Chile. Anion quantifications were performed with Ion Chromatography (Thermo Scientific Dionex ICS-2100), and carbonate speciation was carried out by titration with the Giggenbach method (98). Chemical analyses were validated by the electroneutrality condition of the solution. The admitted ionic balance was ±5% according to the electrical conductivity of the samples (99).

To sample the microbial communities of the mat, approximately 2 mL of the mat was recovered in duplicate with a cork borer, 5 to 20 cm away from the water sample site. Samples were preserved in cryogenic vials with RNAlater reagent (Thermo Fisher Scientific) and stored at −80°C until analysis. DNA extractions were carried out according to Alcorta et al. (100). Briefly, samples were immersed in xanthogenate buffer [1% potassium ethyl xanthogenate (Sigma-Aldrich, USA), 100 mM Tris-HCl (pH 7.4), 20 mM EDTA (pH 8), 800 mM ammonium acetate] and then mechanically shaken (TissueLyzer II, Qiagen) for 1 minute at 30 revolutions per second. Next, samples were incubated for 2 hours at 65°C with 10% SDS. After being mixed in a vortex, the samples were

immersed in ice for 30 minutes. DNA extraction was performed with phenol-chloroform-isoamyl alcohol (25:24:1), while residual phenol was removed with chloroform-isoamyl alcohol (25:1). Nucleic acid precipitation was carried out within 2 hours at −80°C with cold absolute isopropanol and 0.4 M ammonium acetate. The pellet was washed successively with 70% ethanol. The quality and quantity of the recovered nucleic acids were monitored using 1% agarose gel electrophoresis, Qubit (Life Technologies, Carlsbad, California, USA), and Nanodrop (Thermo Fisher Scientific, Waltham Massachusetts, USA). Finally, universal primers 515F (5′-GTGYCAGCMGCCGCGGTAA-3′) and 926R (5′-CCGYCAATTYMTTTRAGTTT-3′) were used to amplify the V4-V5 hypervariable region of the 16S rRNA gene (101). Sequencing of the 22 samples was performed on the Illumina MiSeq platform (Argonne National Laboratory; Lemont, IL, USA). The 16S rRNA sequences of the 11 new samples at El Tatio (in duplicate) were submitted to the NCBI SRA database under the Bioproject accession number PRJNA825489.

## Worldwide hot spring collection data

16S rRNA amplicon sequences from hot springs water (166) and mat (25) samples were retrieved from public databases along with their hydrochemical metadata. The data were obtained from studies where water and 16S rRNA amplicon sequences were sampled in the same field campaign as follows: the initial 758 16S rRNA amplicon samples found were reduced to 414 due to the large heterogeneity of sampling methods and measured variables. A minimum of 3 and a maximum of 25 sampling points per geothermal field were then imposed, randomly removing samples from sub- and over-represented geothermal systems. The final data set consisted of 191 terrestrial hot springs amplicon samples with their respective geographic location, temperature, pH, and concentrations of $Ca^{2+}$, $Mg^{2+}$, $Na^+$, $K^+$, $Cl^-$, $HCO_3^-$, $SO_4^{2-}$, and $SiO_2$. This data set belonged to three previous publications (16, 20, 73) that analyzed 13 geothermal fields (Table 1) distributed over three zones in New Zealand (TVZ), the USA (YPVF), and China (ETPGB). Raw sequences were obtained from the SRA database of the NBCI. The YPVF and the ETPGB metadata were obtained from their respective articles, and the TVZ metadata was recovered from the One Thousand Spring project website (https://1000springs.org.nz/). $HCO_3^-$ concentrations for the YPVF samples were calculated with the USGS geochemical speciation code PHREEQC (102).

## Taxonomic assignation

Each set of sequences was individually imported according to their local source into Qiime2 (103, 104) as demultiplexed sequences. Denoising and resolving the ASVs were performed with the DADA2 pipeline (105) yielding a total of 7,505,257 sequences after filtering (Table 1). At this stage, primers were also removed where necessary. Next, sequences were constrained to the V4 region of the 16S rRNA gene with the cutadapt tool, and then, taxonomic classification was assigned using the q2-feature-classifier (104) against the SILVA-138 99% Operational Taxonomic Unit reference sequences of the V4 region. After removing the mitochondria and chloroplast sequences, a rooted phylogenetic tree was constructed with fasttree2 (106). Counts of each ASV and associated taxonomic tables were exported for subsequent statistical and ecological analyses. For the samples that were in duplicates, those with the highest number of reads were chosen for these analyses.

## Statistical and ecological analysis

An exploratory data analysis was performed on the hydrochemical variables of the 13 geothermal systems. After scaling the data to zero mean and unit variance, Spearman's rank correlation coefficients between these variables were calculated (107). Next, dimensional reduction of the variables was obtained with PCA using the stats package (108).

Sequence analyses were conducted in R with the packages phyloseq (109) and ampvis2 (110). Prior to alpha diversity analyses, rarefaction curves were constructed with the vegan package (111). Community alpha diversity for all samples was obtained using observed richness and Shannon index (112). These indices were grouped and displayed according to habitat (water or mat), study zone, and geothermal field. They were then tested for statistical significance using the Kruskal-Wallis test (113). Spearman's correlation index between richness and hydrochemical variables was calculated for all samples and by study zone. The significant and highest correlations between chemical parameters and alpha diversity in each study zone were then plotted in two-component diagrams.

Prior to beta diversity analyses, read counts were normalized using the DESeq2 package (114). An MDS was performed based on weighted Unifrac distances (115) which were then used in a PERMANOVA (116) using the adonis2 function of the vegan package with 9,999 permutations using observed richness, habitat (mat or water), and PCA eigenvectors as explanatory variables.

In addition, to assess the effect of hydrochemical gradients on selected taxa, Spearman's correlation index was calculated between physicochemical parameters and taxa whose relative abundance in their study zone exceeded 0.1% at the phylum and family levels. Only taxa with significant correlations ($P < 0.05$) greater than 0.3 were selected and plotted on a heatmap. Then, microbial co-occurrence networks were performed per study zone with the Sparse Inverse Covariance Estimation for Ecological Association Inference method (117) and plotted with ggnet (118).

Finally, a distance decay analysis was conducted based on similarity/dissimilarity indexes: Jaccard (119), Bray-Curtis (120), unweighted Unifrac (121), and weighted Unifrac (115). Geodesic distances between samples were obtained through the geosphere package (122), and then, these distances were used to define spatial scales (local: 0–58 km; regional: 58–279 km; global: 280–8,576 km). A linear regression on the logarithmic scale was performed for the four indexes and the geodesic distances between samples at each of the three spatial scales, and then, their $R^2$, $P$-value, and slope were used for further discussion.

## ACKNOWLEDGMENTS

The authors acknowledge Johanna Saldías for her logistical support in the BD Lab during this research and Verónica Rodríguez for helping with the hydrochemical analyses. The authors also acknowledge Caspana and Toconce Aymara Communities for the permission to the fieldwork performed at El Tatio geothermal field.

This work was financially supported by the Chilean National Agency of Research and Development (ANID) through the Andean Geothermal Center of Excellence (CEGA, ANID-FONDAP 15090013, 15200001, and ACE210005), Millennium Institute Center for Genome Regulation (ANID – Millennium Science Initiative Program – ICN2021_044), FONDECYT 1190998 and 1230217, Center for Climate and Resilience Research (CR)2 (ANID-FONDAP 1522A0001), a National Master Scholarship granted to C.B. (22200311), and a National Doctoral Scholarship granted to J.T.L. (21171048) during the analysis and writing of this manuscript. Additional support was provided by the research effort Iniciativa de Investigación UnACh 2020-132-Unach.

## AUTHOR AFFILIATIONS

[1]Department of Geology, University of Chile, Santiago, Chile
[2]Andean Geothermal Center of Excellence (CEGA-Fondap), University of Chile, Santiago, Chile
[3]Department of Molecular Genetics and Microbiology, Pontifical Catholic University of Chile, Santiago, Chile
[4]Center for Climate and Resilience Research (CR)2, University of Chile, Santiago, Chile
[5]Millennium Institute Center of Genome Regulation (CGR), Santiago, Chile

[6]Laboratorio de Bioinformática, Facultad de Educación, Universidad Adventista de Chile, Chillán, Chile

## AUTHOR ORCIDs

Carla Barbosa  http://orcid.org/0000-0001-5682-9959
Jaime Alcorta  http://orcid.org/0000-0001-7662-239X
Beatríz Díez  http://orcid.org/0000-0002-9371-8083

## FUNDING

| Funder | Grant(s) | Author(s) |
|---|---|---|
| Agencia Nacional de Investigación y Desarrollo (ANID) | Beca de Magister Nacional 22200311 | Carla Barbosa |
| Agencia Nacional de Investigación y Desarrollo (ANID) | FONDECYT 1190998 | Beatríz Díez |
| Agencia Nacional de Investigación y Desarrollo (ANID) | CEGA ANID-FONDAP 15090013 15200001 & ACE210005 | Diego Morata |
| Millennium Institute Center for Genome Regulation | ICN2021_044 | Beatríz Díez Jaime Alcorta Javier Tamayo-Leiva |
| Universidad Adventista de Chile | Iniciativa de Investigación UnACh 2020-132- 701 Unach | Oscar Salgado |
| Agencia Nacional de Investigación y Desarrollo (ANID) | Beca de Doctorado Nacional 21171048 | Javier Tamayo-Leiva |
| Agencia Nacional de Investigación y Desarrollo | FONDECYT 1230217 | |

## AUTHOR CONTRIBUTIONS

Carla Barbosa, Conceptualization, Formal analysis, Methodology, Writing – original draft, Writing – review and editing | Javier Tamayo-Leiva, Conceptualization, Formal analysis, Methodology, Writing – original draft, Writing – review and editing | Jaime Alcorta, Conceptualization, Methodology, Writing – original draft, Writing – review and editing | Oscar Salgado, Funding acquisition, Methodology | Linda Daniele, Writing – review and editing | Diego Morata, Conceptualization, Funding acquisition, Resources, Supervision.

## DATA AVAILABILITY

The 16S rRNA sequences of the 11 new samples at El Tatio (in duplicate) were submitted to the NCBI Sequence Read Archive (SRA) database under the Bioproject accession number PRJNA825489.

## ADDITIONAL FILES

The following material is available online.

### Supplemental Material

**Figures S1 to S11, Tables S1 to S4, Text S1 (Spectrum00249-23-s0001.docx).** The file contains all the supplementary material of the manuscript

### Open Peer Review

**PEER REVIEW HISTORY (review-history.pdf).** An accounting of the reviewer comments and feedback.

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
