## [Reviewer comments · Microbiology Spectrum]

Microbiology Spectrum

Effects of hydrogeochemistry on the microbial ecology of terrestrial hot springs

Carla Barbosa, Javier Tamayo-Leiva, Jaime Alcorta, Oscar Salgado, Linda Daniele, Diego Morata, and Beatriz Díez

Corresponding Author(s): Beatriz Díez, Pontificia Universidad Católica de Chile

Review Timeline:

Submission Date:	January 19, 2023
Editorial Decision:	February 26, 2023
Revision Received:	April 24, 2023
Editorial Decision:	June 2, 2023
Revision Received:	June 19, 2023
Accepted:	July 13, 2023

Editor: Sandi Orlic

Reviewer(s): The reviewers have opted to remain anonymous.

Transaction Report:

DOI: <https://doi.org/10.1128/spectrum.00249-23>

February 26, 2023

Dr. Beatriz Diez Moreno
Pontificia Universidad Católica de Chile
Santiago
Chile

Re: Spectrum00249-23 (Effects of hydrogeochemistry on the microbial ecology of terrestrial hot springs)

Dear Dr. Beatriz Diez Moreno:

Thank you for submitting your manuscript to Microbiology Spectrum. As you will see your paper is very close to acceptance. Please modify the manuscript along the lines I have recommended. As these revisions are quite minor, I expect that you should be able to turn in the revised paper in less than 30 days, if not sooner. If your manuscript was reviewed, you will find the reviewers' comments below.

When submitting the revised version of your paper, please provide (1) point-by-point responses to the issues raised by the reviewers as file type "Response to Reviewers," not in your cover letter, and (2) a PDF file that indicates the changes from the original submission (by highlighting or underlining the changes) as file type "Marked Up Manuscript - For Review Only". Please use this link to submit your revised manuscript. Detailed instructions on submitting your revised paper are below.

Link Not Available

Sincerely,

Sandi Orlic

Reviewer comments:

Reviewer #1 (Comments for the Author):

I am unable to paste my comment here because the system tells me I cannot use the word accepted or rejected even though I don't use them. Please copy my entire comment to the editor and send them to the authors

Reviewer #2 (Comments for the Author):

General comments:

This manuscript compares microbial communities in hot springs found in New Zealand, Yellowstone, and the Tibetan Plateau to identify global trends in primary physico-chemical conditions that control microbial distributions. Overall, this is a well-written manuscript that is able to combine both new data with previously published sequence data to derive new understanding. While this manuscript has significant merit, I would suggest a few things that will help streamline the narrative to fit with the readership of this journal and emphasize the key finds more. I fully appreciate the thorough description of the geology, but in many places it sidetracks from the main message. For example, while the deep processes affect the surface chemistry, it is only the surface chemistry that these microbes would be responding to. Section 2.1 provides the necessary background and description of the environment. In further sections that explain why the chemistry is that way is not needed. I would also suggest having a conclusions section at the end of the results and discussion. There is a lot of great information thrown at the reader but nothing to tie it all together.

Specific comments:

Line 71: These can also be more alkaline. I would say neutral to alkaline pH

Line 222: The total number of samples are mentioned later, but this is the first reference to it and it would be good to add that number here.

Figure 2.6: It is difficult to keep the different parts of the figure separate. I suggest removing the grid and adding a clearly defined line for each axis. Also, all figures numbers should be updated to be 1,2, 3, etc.

Line 357: The rest of the phyla names mentioned are the updated phyla name, except for Proteobacteria (Pseudomonadota) and Cyanobacteria (Cyanobacteriota). I know the utility of this is debated and not all databases have updated to these. I am only pointing this out in case these names were missed. Otherwise leave them as is.

Line 383: I took be reading below to figure out what PC4 represents. A little definition here would be helpful.

Line 420: "courses" can be removed and make drainage plural.

Line 439: remove "the"

Line 475-477: Here is an example of what I was mentioning in the general comments.

Line 548-549: It would be helpful to have a little more information on how these samples were selected. For example, what was the requirements for sampling and DNA extraction procedures. Were they all sequenced using the same method?

Line 599-601: I don't believe co-occurrence data was presented in the results. I would like to see this and help support the conclusion that other variables such as biotic interactions help to structure the community.

The supplemental material should be referenced in the main manuscript where it is most appropriate.

Preparing Revision Guidelines

Please return the manuscript within 60 days; if you cannot complete the modification within this time period, please contact me. If you do not wish to modify the manuscript and prefer to submit it to another journal, please notify me of your decision immediately so that the manuscript may be formally withdrawn from consideration by Microbiology Spectrum.

Corresponding authors may join or renew ASM membership to obtain discounts on publication fees. Need to upgrade your

membership level? Please contact Customer Service at Service@asmusa.org.

Dear Microbiology Spectrum Journal Editors,

We appreciate all reviewers' comments which helped to improve the quality of our manuscript. In this new version, we considered the suggestions of the reviewers to discuss how comparable are the four datasets and how besides this, the obtained information contributes to the scientific community. Here you will find the one-by-one responses to the criticisms of the reviewers.

Reviewer #1

General comments

This study focuses on the geochemistry and microbial diversity - as observed via 16S rRNA gene amplicon sequencing - of 11 newly sampled Chilean hot springs as well as a couple hundred published 16S rRNA gene amplicon datasets obtained from geothermal features in China, New Zealand, and the US. Overall this is a very descriptive, observational study that doesn't claim to be anything more than correlative observations of different geochemical parameters and certain microbial taxonomies being observed. Overall, I believe this study falls within a category of studies specifically sought-for by the journal, i.e. "Descriptive datasets that would serve as a community resource"; that phrase describes this study well.

That said, I have a few foundational concerns about how the study was designed that need to be addressed before I can begin to properly evaluate it. It is unclear to me whether the newly obtained and studied samples (11) can be properly compared with those from the literature - or whether those from the literature can be directly compared-, because it is unclear whether the methodology employed by those studies and whether the samples retrieved by them can actually be compared. This caution applies three-ways:

(i) It is important to consider how water and DNA samples were taken in different studies. Every study measured water chemistry and obtained DNA samples. However, it is unclear if it's not apples and oranges that are compared here. For example, mat and sediment samples were taken for DNA analyses but did some of the studies look at pore water chemistry rather than the chemistry of the overlying water? If the latter, how far away from the site of DNA sampling does the water come from? Can pore water data and overlying water even be compared?

R: Authors appreciate the concerns of the reviewers, which helped to improve the manuscript. As suggested, this study was intended to be a resource for the scientific community in terms of global comparisons of hydrogeochemical analyses, to determine the effect of the former on the structure of microbial assemblages at regional scales. Furthermore, one important point of our study is how geological processes in the four different regions can influence the microbial communities along with hydrogeochemistry as an intermediary factor, therefore this is a step forward to connect both phenomena at wider scales. We acknowledge that the methods allow for comparisons that have to be taken with caution, however, the analyses show a significant effect of hydrogeochemistry, and explain a considerable part of the variance in the microbial communities in each zone. It is worth mentioning that there is a plethora of 16S rRNA studies of hot springs microbial communities, however, we applied a number of constraints to carefully choose the

geographical regions, measured parameters, amplified hypervariable regions, and sample size. This latter is mentioned in lines 557 - 562.

As for the data of all the studies used, including our own dataset, they did not consider pore water samples, but only overlying water from the hot springs. Thus, we think that there is no need to further discuss how comparable pore water and overlying water are. This was clarified in lines 482- 483.

On the other hand, water from the El Tatio hot springs was sampled at 5 to 20 centimeters from the microbial mats, which was now added in lines 535 - 536. From the methodology of Power et al. (2018), it follows that water for chemical analysis in the Taupo Volcanic Zone was sampled from the same site as the microbial communities (3 L of spring water from the center or 3 m away from the center of the features for both DNA and geochemical analyses). On the contrary, detailed information on the distance between microbial mat samples and water samples in the Yellowstone Plateau Volcanic Field is not provided in Hamilton et al. (2019). The methodology of Guo et al. (2020) also does not mention at what distances water samples for DNA extractions were taken in comparison to the water for hydrochemical analysis on the Eastern Tibetan Plateau. Thus, the authors recognize that this may represent a methodological bias, which is mentioned now in lines 485 - 486.

(ii) It is unclear whether the methods to determine different chemical parameters (e.g., ions) can be directly compared to each other. Do they have varying accuracy and sensitivity, and if so, does it matter for interpreting the results obtained by different methods?

R: Temperature and pH in El Tatio were measured with a multiparameter analyzer WTW Multi340i, with the accuracy of 0.1°C and 0.01, respectively. This information was added to the text in lines 521 - 523. From the data collected, the accuracy of the digital thermometer used for samples in the Eastern Tibetan Plateau was explicitly given by Guo et al. (2020) (0.1°C), as well as the pH precision (0.01). The multiparameter analyzer used in Yellowstone Plateau Volcanic Field was a WTW Multi330i, which also have temperature and pH accuracies of 0.1°C and 0.01, respectively. In contrast, the temperature of samples in the Taupo Volcanic Zone was measured with a Fluke 51-II thermocouple with an accuracy of 0.3°C and no detailed information about the multiparameter analyzer used for pH measurements was given by Power et al. (2018). In consequence, comparing these sets of data could lead to a bias in temperature and pH measurements. This is now discussed in lines 484 – 484.

Methods for chemical analysis are mostly different for each study, as Supplementary Table 1 shows. However, this fact should not be a limitation for the comparison of major elements, which are mostly present in high concentrations compared to the detection limits and sensitivity values of the equipment. This is now discussed in lines 487 – 490. This may take special relevance for analyzing minor and trace elements, where accuracy, sensitivity, and detection limit differences between datasets should be evaluated.

(iii) Published datasets used very different primer sets, making it very hard to draw comparisons between published datasets and the new datasets. For example, you use primers 515F-926R, while the largest published dataset (Power2018) that was reanalyzed by you used 515F-806R. 515F-926R covers approximately 81% of archaea, 85% of bacteria and 81% of eukaryotic SSU rRNA sequences, according got TestPrime. In contrast, 515F-

806R covers only 50% of archaea and 8.5% of bacteria, and no eukaryotic sequences, with varying coverage of specific phyla. It is unclear to me whether we can compare these datasets because our view into microbial community structure is flawed.

R:

Authors acknowledge this valuable comment. It is known that the primer sets and even the PCR conditions can target differentially the microbial communities. To address this issue, we chose only studies that target at least the V4 region, which is covered by the primer set we used (V4-V5). As mentioned above, we carefully selected studies with extant hydrogeochemical data, a bigger sample size, and the same hypervariable region amplified as said in lines 557 - 562. However, the reviewer is right in the issue that besides targeting the same hypervariable region (therefore having the same primer name, e.g., 515F), the used primers had different lengths and ambiguous bases which showed differential proportions of matched sequences with the SSU Silva r138.1 database according to TestPrime. All forward primers targeted the 515 region, while the three previous studies used the 806R region for the reverse primer (our study used the 926R primer as noted by the reviewer). Therefore, as suggested, we performed the suggested analyses and the TestPrime results showed (as also highlighted by reviewer) that the Power et al. (2018) primers matched 9.3 % of the sequences, Hamilton et al. (2019) matched 71.4 %, Guo et al. (2021) matched 72.4 % and our primers matched 83.9 % of the NR silva sequences. While this may be a drawback, we further observed the coverage values for specific taxa that are highlighted in the text, where we found similar coverage values between the four primer sets for phyla like Thermoplasmatota (Archaea), Aquificota, Deinococcota, and Thermotoga. By other side, similar values for three sets excepting the Power et al. (2018) primers, which were lower, were found for Armatimonadota (~85 % vs 57), Bacteroidota (~86 % vs 1.2), Chloroflexi (~60 % vs 32), Cyanobacteria (~80% vs 25) and Proteobacteria (~88 % vs 1.5). Therefore, we can suggest amplification bias for some phyla. However, we also found a high representation of proteobacteria in the TVZ (Power et al. 2018) besides their low primer coverage, which could show that the bias seen by in silico results could not be specifically represented by the in vitro conditions. Taking all of this into consideration, the differential potential amplification of the microbial communities due to the primer sets is now discussed in lines 490 – 498. Furthermore, as our comparative analysis is between different studies and regions, we suggest that the microbial communities are still comparable inside each region regarding their own physicochemical parameters and amplification biased, as well between sites, as the hot spring microbial communities tend to be less diverse than other environments.

Minor comments

L513, how close to the DNA samples was the water sample taken from? How can you be sure that the water chemistry reflects the chemistry in the mat.

R: The distance between DNA and water samples in El Tatio ranges from 5 to 20 centimeters, which was added in lines 535 - 536. Water chemical analyses do not accurately reflect the chemistry in the mat, which probably shows small-scale hydrochemical gradients.

Instead, these analyses aim to provide a general description of each hot spring under study in terms of major elements, considering the objective and scale of the study.

L390-392, you could test this hypothesis by only considering samples >75°C in temperature, i.e., above the temperature at which photosynthetic pigments break down.

R: We acknowledge this idea and PERMANOVA results were compared between samples at temperatures below 75°C and over 75 °C. Results agreed with this statement, and now were incorporated in lines 348 – 351 and Supplementary Table 3. Briefly, the microbial community structure at temperatures below 75°C was explained in 11% by statistically significant correlations ($p < 0.001$) related to temperature, pH, and hydrochemistry. On the other hand, the structure of the microbial communities at temperatures over 75°C was explained in 26% by statistically significant correlations ($p < 0.01$) related to temperature, pH, and hydrochemistry. This result is consistent with the hypothesis cited and authors appreciate the suggestion of the reviewer.

Reviewer #2

General comments

This manuscript compares microbial communities in hot springs found in New Zealand, Yellowstone, and the Tibetan Plateau to identify global trends in primary physico-chemical conditions that control microbial distributions. Overall, this is a well-written manuscript that is able to combine both new data with previously published sequence data to derive new understanding.

While this manuscript has significant merit, I would suggest a few things that will help streamline the narrative to fit with the readership of this journal and emphasize the key finds more. I fully appreciate the thorough description of the geology, but in many places it sidetracks from the main message. For example, while the deep processes affect the surface chemistry, it is only the surface chemistry that these microbes would be responding to. Section 2.1 provides the necessary background and description of the environment. In further sections that explain why the chemistry is that way is not needed.

I would also suggest having a conclusions section at the end of the results and discussion. There is a lot of great information thrown at the reader but nothing to tie it all together.

R: We appreciate the feedback provided by reviewer #2. We agree with the concern about the length of the geological descriptions and geochemical interpretations and therefore we shortened the section 'Hot springs hydrochemistry'. We left the full text in Supplementary Text 1 in case it would be helpful for readers. One of the main aims of this study was to determine potential correlations between geothermal processes and the microbial community assemblages, therefore we think these descriptions contribute to this purpose.

Additionally, in this new version of the manuscript, we also provide a final section that contained the most relevant contributions of this study (lines 500 - 517). Here, is the added text:

Terrestrial hot springs are a valuable natural resource that host an exceptional biotic diversity. This study attempted to compare the microbial diversity and hydrochemistry of 202 hot springs from 4 geothermal areas of the world. The comparison of data from different studies revealed multiple biases, mainly associated with the methodologies of analytical chemistry, nucleic acids extraction and amplification. Despite these limitations, statistical analysis showed that 24.4% of the variance in thermal community structure was explained by temperature, pH and hydrochemistry of major elements. This highlights the relevance of these parameters in the microbial ecology of thermophilic communities and the potential influence of geochemical processes on them. At the same time, hydrogeochemistry was consistent with the types of underlying geological processes, showing their influence on the assembly of microbial communities. Furthermore, correlations of dominant taxa (such as Chloroflexota, Bacteroidota, Cyanobacteria, Deinococcota, Thermoplasmatota, Aquificota, and Armatimonadota) with hydrogeochemistry showed that these processes also influence their biotic interaction in microbial communities. Meanwhile, these assemblages showed consistency across the globe as their microbes appear to diverge early in the same taxonomic groups between globally distant thermal zones. Altogether, these results indicate the need to develop collaborative and standardized global efforts to better understand the effect of hydrochemistry on microbial communities, as well as to continue to conduct local studies of geothermal fields located in different areas of the globe, taking into account the singularities of each zone and its particular microbial communities.

Minor comments

Line 71: These can also be more alkaline. I would say neutral to alkaline pH

R: Authors appreciate this observation, and the text was changed as suggested (line 69).

Line 222: The total number of samples are mentioned later, but this is the first reference to it and it would be good to add that number here.

R: Authors appreciate this observation and was taken into consideration in the manuscript (line 221).

Figure 2.6: It is difficult to keep the different parts of the figure separate. I suggest removing the grid and adding a clearly defined line for each axis. Also, all figures numbers should be updated to be 1,2, 3, etc.

R: Authors appreciate this comment and added a defined line for each axis of the graphs. However, we suggest keeping the grids that help the readers to easily realize that figures A and B share the same Axis X, and figures A and C share the same Axis Y.

In addition, the numbering of the figures has been corrected as requested.

Line 357: The rest of the phyla names mentioned are the updated phyla name, except for Proteobacteria (Pseudomonadota) and Cyanobacteria (Cyanobacteridota). I know the utility

of this is debated and not all databases have updated to these. I am only pointing this out in case these names were missed. Otherwise leave them as is.

R: The database which was used to assign the taxonomy to the ASVs was the SILVA-138 and it is not updated in those specific phyla. We agree with the reviewer that it is a topic still in debate and we prefer to maintain consistency with the database.

Line 383: I took be reading below to figure out what PC4 represents. A little definition here would be helpful.

R: Authors appreciate this observation, and a brief definition was added in line 341.

Line 420: "courses" can be removed and make drainage plural.

R: The text was changed accordingly (line 377).

Line 439: remove "the"

R: The text was changed accordingly (line 396).

Line 475-477: Here is an example of what I was mentioning in the general comments.

R: Authors appreciate this observation. However, this is exactly one of the examples of how deep processes would influence hot spring microbial communities. The precipitation of the mineral phases mentioned in line 446 would prevent or reduce the formation of acid-sulfate water and, in consequence, would limit the presence of acidophilic microbial communities, which is now explicit in lines 447 – 449. We think that this is an example where hydrothermal and geothermal processes impact the microbial communities that inhabit in terrestrial hot springs.

Line 548-549: It would be helpful to have a little more information on how these samples were selected. For example, what was the requirements for sampling and DNA extraction procedures. Were they all sequenced using the same method?

R: Since there are a plethora of 16S rRNA studies from microbial communities from hot springs, we applied a number of constraints to carefully choose the geographical regions, measured parameters, amplified hypervariable regions, and sample size. This is mentioned in lines 557 - 562. Samples were selected from studies where water and 16S rRNA amplicon sequences were sampled in the same field campaign, which was added in line 558. As Supplementary Table 1 shows, all sequences belong to the V4 region of the 16S rRNA gene. However, DNA extraction (all different) and sequencing methods (one Ion Torrent and three Illumina MiSeq methodologies) were not the same for all samples, which is also included now in Supplementary Table 4.

Line 599-601: I don't believe co-occurrence data was presented in the results. I would like to see this and help support the conclusion that other variables such as biotic interactions help to structure the community.

R: Authors appreciate this observation. The co-occurrence data was not included in the first version of the manuscript, to reduce the words of the main text. However, due to the reviewer's appreciation, the main results of this analysis are now included in lines 397 - 407.

The supplemental material should be referenced in the main manuscript where it is most appropriate.

R: Authors appreciate this comment, and the Supplementary Material is referenced where most appropriate.

June 2, 2023

Prof. Beatriz Díez
Pontificia Universidad Católica de Chile
Department of Molecular Genetics and Microbiology
Santiago
Chile

Re: Spectrum00249-23R1 (Effects of hydrogeochemistry on the microbial ecology of terrestrial hot springs)

Dear Prof. Beatriz Díez:

Thank you for submitting your manuscript to Microbiology Spectrum. As you will see your paper is very close to acceptance. Please modify the manuscript along the lines I have recommended. As these revisions are quite minor, I expect that you should be able to turn in the revised paper in less than 30 days, if not sooner. If your manuscript was reviewed, you will find the reviewers' comments below.

When submitting the revised version of your paper, please provide (1) point-by-point responses to the issues raised by the reviewers as file type "Response to Reviewers," not in your cover letter, and (2) a PDF file that indicates the changes from the original submission (by highlighting or underlining the changes) as file type "Marked Up Manuscript - For Review Only". Please use this link to submit your revised manuscript. Detailed instructions on submitting your revised paper are below.

Link Not Available

Sincerely,

Sandi Orlic

Reviewer comments:

Reviewer #1 (Comments for the Author):

In my opinion, the methodological details that are not reported in the original papers that generated the data reported by the authors substantially limit what statements can now be made in this new study. This is not the author's fault - nevertheless, it doesn't change the fact that some statement can simply not be made because it is unclear - as the authors themselves state - how geochemistry vs. DNA samples in the original papers were obtained. In my opinion, those original papers should have never gone through peer review without those inconsistencies having been clarified. I believe the authors of this study have rephrased the manuscript adequately to draw the attention of the reader to these potential drawbacks.

I thank the authors for detailing methodological differences in how geochemical parameters were taken. I agree with them that differences in methodologies should have not lead to difference of outcome.

The primer issue, however, is still a fundamental problem that has not been addressed. I believe there's, however, a way to reconcile it. As you state, all primers cover the V4 region. The most pragmatic thing to do is trim all sequences that were generated using the 515F-926R primerset and trim the sequences down to the 515F-806R region. If other primers were used the same needs to be done; the same region should be analyzed because each primer set - as detailed in my original review and reaffirmed by the authors rebuttal - targets different phyla differently. Also, a superficial view at how different phyla are targeted by different primers is not enough - phyla are the highest taxonomic rank under domain and encompass a massive diversity of organisms; saying that one primerset covers ~85% of Armatimonadota and the other 57 of them is only side of the coin - are the 57% targeted by primerset 2 all targeted by the first 1? This needs to be considered for all taxa and all ranks.

Obviously, this is not feasible! The only solution to the problem is to compare apples to apples. The solution to redo the analyses and trim all sequences down to the same length and the same region - that way, even though not all taxa are covered, at least all samples, independent of who generated them, are equally affected.

I believe that an analysis comparing data that were generated using different primer sets that have massively different coverage and then explicitly comparing samples that were targeted with said primers with each other is of insufficient quality and hardly any conclusions can be drawn from such an analysis. Unless the problem has been addressed, this study not be published because it doesn't fulfill the criteria I expect from a Microbiology Spectrum publication.

I understand this sounds like a lot of work because I am effectively asking you to redo the entire analyses. However, I suggest that you see this as a change. Redo the analyses and then report differences and similarities between the two ways you did the analyses. That way you generate a very valuable benchmarking paper comparing the two ways of doing things. Independent of what they outcome would be - the two ways of analyses largely agree or they don't - it would be a very valuable contribution to the scientific literature that might much wider cited than just by researchers in the field of thermal biology because you would be setting a precedence that would be widely cited!

Reviewer #2 (Comments for the Author):

All of my comments have been appropriately modified.

Preparing Revision Guidelines

Please return the manuscript within 60 days; if you cannot complete the modification within this time period, please contact me. If you do not wish to modify the manuscript and prefer to submit it to another journal, please notify me of your decision immediately so that the manuscript may be formally withdrawn from consideration by Microbiology Spectrum.

Dear Microbiology Spectrum Journal Editors,

We appreciate the reviewers' comments. We acknowledge the methodological discussion, which enabled us to explicitly address certain pitfalls when comparing data from multiple studies. Additionally, we drew significant conclusions regarding the influence of geochemistry on microbial communities.

Reviewer #1

In my opinion, the methodological details that are not reported in the original papers that generated the data reported by the authors substantially limit what statements can now be made in this new study. This is not the author's fault - nevertheless, it doesn't change the fact that some statements can simply not be made because it is unclear - as the authors themselves state - how geochemistry vs. DNA samples in the original papers were obtained. In my opinion, those original papers should have never gone through peer review without those inconsistencies having been clarified. I believe the authors of this study have rephrased the manuscript adequately to draw the attention of the reader to these potential drawbacks.

I thank the authors for detailing methodological differences in how geochemical parameters were taken. I agree with them that differences in methodologies should have not lead to difference of outcome.

R: Authors appreciate these valuable comments as they enable us to provide more explicit details about our methods, and they will serve as an example to follow in future studies. Furthermore, a common agreement was reached with the reviewer regarding the comparability of our results despite methodological differences.

The primer issue, however, is still a fundamental problem that has not been addressed. I believe there's, however, a way to reconcile it. As you state, all primers cover the V4 region. The most pragmatic thing to do is trim all sequences that were generated using the 515F-926R primerset and trim the sequences down to the 515F-806R region. If other primers were used the same needs to be done; the same region should be analyzed because each primer set - as detailed in my original review and reaffirmed by the authors rebuttal - targets different phyla differently. Also, a superficial view at how different phyla are targeted by different primers is not enough - phyla are the highest taxonomic rank under domain and encompass a massive diversity of organisms; saying that one primerset covers ~85% of Armatimonadota and the other 57 of them is only side of the coin - are the 57% targeted by primerset 2 all targeted by the first 1? This needs to be considered for all taxa and all ranks. Obviously, this is not feasible! The only solution to the problem is to compare apples to apples. The solution to redo the analyses and trim all sequences down to the same length and the same region - that way, even though not all taxa are covered, at least all samples, independent of who generated them, are equally affected.

I believe that an analysis comparing data that were generated using different primer sets that have massively different coverage and then explicitly comparing samples that were targeted with said primers with each other is of insufficient quality and hardly any conclusions can be drawn from such an analysis. Unless the problem has been addressed, this study not be published because it doesn't fulfill the criteria I expect from a Microbiology Spectrum publication.

I understand this sounds like a lot of work because I am effectively asking you to redo the entire analyses. However, I suggest that you see this as a change. Redo the analyses and then report differences and similarities between the two ways you did the analyses. That way you generate a very valuable benchmarking paper comparing the two ways of doing things. Independent of what they outcome would be - the two ways of analyses largely agree or they don't - it would be a very valuable contribution to the scientific literature that might much wider cited than just by researchers in the field of thermal biology because you would be setting a precedence that would be widely cited!

R: Authors understand the concerns of the reviewer regarding the differences in the primer coverage at lower taxonomic levels. It is understood that different primers can introduce biases, but it should be noted that new primers are continuously being developed to target a broader range of diversity. Additionally, it is common for studies conducted in different years to employ different primer sets, as demonstrated in the present study. For example, Merkel et al. (2019) investigated different primer sets to profiling hot spring microbial communities, but their findings did not include results for any of the primers utilized in our study (doi: 10.1134/S0026261719060110). Our selection of the 515F - 926R primer pair was based on the work of Parada et al. (2016), who found that this pair yielded a closer representation of the original sample's community composition compared to other primer pairs (doi:10.1111/1462-2920.13023). Furthermore, this primer pair is currently included in the Earth Microbiome project's protocol, enabling comparisons with other environments. Once again, we appreciate the discussion surrounding this matter.

Regarding the suggested analysis, we think that the proposed protocol suggested by the reviewer aligns with the approach we have taken. In lines 575 - 578 we stated that "Next, sequences were constrained to the V4 region of the 16S rRNA gene with the cutadapt tool and then taxonomic classification was assigned using the q2-feature-classifier (100) against the SILVA-138 99% Operational Taxonomic Unit (OTU) reference sequences of the V4 region". Therefore, besides our primers targeting the V4-V5 region, we only used the common V4 region for all samples included in our study. While we believe our methods are adequately described, we have taken your suggestion into account and added a statement in the results section (lines 264 - 265) to further emphasize this point for the readers.

We don't have analyses performed with the entire V4-V5 region of our amplicon samples and classified separately from the V4 regions from the other studies. Consequently, the requested comparison is not available, as we initially conducted the methodology as requested by the reviewer. If necessary, these analyses can be performed, but we believe it is common practice to use a shared region for comparative analyses between studies or samples. Once again, we acknowledge the valuable comments provided by the reviewer.

Reviewer #2

All of my comments have been appropriately modified.

R: Authors appreciate the comments from the reviewer that helped to improve the manuscript.

July 10, 2023

Prof. Beatriz Díez
Pontificia Universidad Católica de Chile
Department of Molecular Genetics and Microbiology
Santiago
Chile

Re: Spectrum00249-23R2 (Effects of hydrogeochemistry on the microbial ecology of terrestrial hot springs)

Dear Prof. Beatriz Díez:

Your manuscript has been accepted, and I am forwarding it to the ASM Journals Department for publication. You will be notified when your proofs are ready to be viewed.

Sincerely,

Sandi Orlic
Editor, Microbiology Spectrum
